# A Simple and Scalable Representation for Graph Generation

**Yunhui Jang**[1]**, Seul Lee**[2]**, Sungsoo Ahn**[1]
[1]Pohang University of Science and Technology
[2]Korea Advanced Institute of Science and Technology
{uni5510,sungsoo.ahn}@postech.ac.kr, seul.lee@kaist.ac.kr

## Abstract

Recently, there has been a surge of interest in employing neural networks for graph generation, a fundamental statistical learning problem with critical applications like molecule design and community analysis. However, most approaches encounter significant limitations when generating large-scale graphs. This is due to their requirement to output the full adjacency matrices whose size grows quadratically with the number of nodes. In response to this challenge, we introduce a new, simple, and scalable graph representation named gap encoded edge list (GEEL) that has a small representation size that aligns with the number of edges. In addition, GEEL significantly reduces the vocabulary size by incorporating the gap encoding and bandwidth restriction schemes. GEEL can be autoregressively generated with the incorporation of node positional encoding, and we further extend GEEL to deal with attributed graphs by designing a new grammar. Our findings reveal that the adoption of this compact representation not only enhances scalability but also bolsters performance by simplifying the graph generation process. We conduct a comprehensive evaluation across ten non-attributed and two molecular graph generation tasks, demonstrating the effectiveness of GEEL.

## 1 Introduction

Learning the distribution over graphs is a challenging problem across various domains, including social network analysis (Grover et al., 2019) and molecular design (Li et al., 2018; Maziarka et al., 2020). Recently, neural networks gained much attention in addressing this challenge by leveraging the advancements in deep generative models, e.g., diffusion models (Ho et al., 2020), to show promising results. These works are further categorized based on the graph representations they employ.

However, the majority of the graph generative models do not scale to large graphs, since they generate the adjacency matrix-based graph representations (Simonovsky & Komodakis, 2018; Madhawa et al., 2019; Liu et al., 2021; Maziarka et al., 2020). In particular, for large graphs with $N$ nodes, the adjacency matrix is hard to handle since they consist of $N^2$ binary elements. For example, employing a Transformer-based autoregressive model for all the binary elements requires $O(N^4)$ computational complexity. Researchers have considered tree-based (Segler et al., 2018) or motif-based representations (Jin et al., 2018; 2020) to mitigate this issue, but these representations constrain the graphs being generated, e.g., molecules or graphs with motifs extracted from training data.

Intriguingly, a few works (Goyal et al., 2020; Bacciu & Podda, 2021) have considered generating the edge list representations as a potential solution for large-scale graph generation. In particular, the list contains $M$ edges that are fewer than $N^2$ elements in the adjacency matrix, a distinctive difference especially for sparse graphs. However, such edge list-based graph generative models instead suffer from the vast vocabulary size $N^2$ for the possible edges. Consequently, they face the challenge of learning dependencies over a larger output space and may overfit to a specific edge or an edge combination appearing only in a few samples. Indeed, the edge list-based representations empirically perform even worse than simple adjacency matrix-based models (You et al., 2018), e.g., see Table 1.

In this paper, we propose a simple, scalable, yet effective graph representation for graph generation, coined **G**ap **E**ncoded **E**dge **L**ist (GEEL). On one hand, grounded in edge lists, GEEL enjoys a

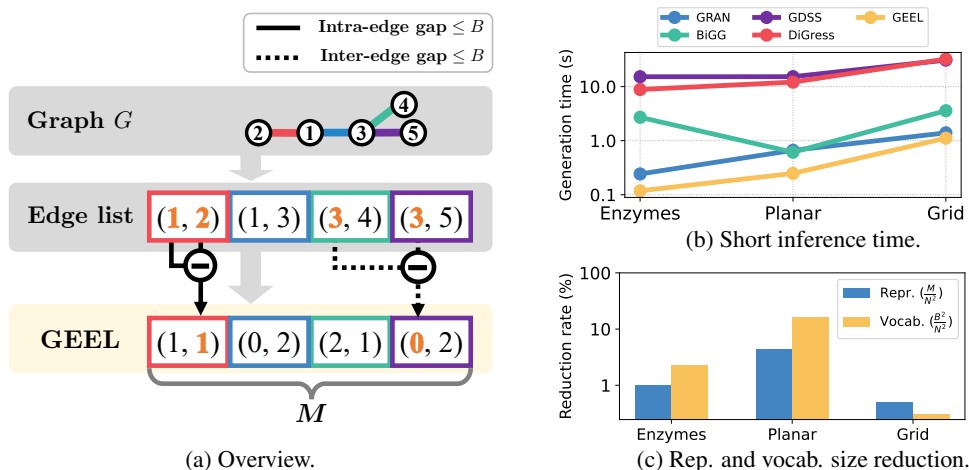

(a) Overview.

(b) Short inference time.

(c) Rep. and vocab. size reduction.

Figure 1: **Overview and advantages of gap encoded edge list (GEEL)**.

compact representation size that aligns with the number of edges. On the other hand, GEEL improves the edge list representations by significantly reducing the vocabulary size with gap encodings that replace the node indices with the difference between nodes, i.e., gap, as described in Figure 1a. We also promote bandwidth restriction (Diamant et al., 2023) which further reduces the vocabulary size. Next, we augment the GEEL generation with node positional encoding. Finally, we introduce a new grammar for the extension of GEEL to attributed graphs.

The advantages of our GEEL are primarily twofold: scalability and efficacy. First, regarding scalability, the reduced representation and the vocabulary sizes mitigate the computational and memory complexity, especially for sparse graphs, as described in Figure 1b. Second, concerning the efficacy, GEEL narrows down the search space to $B^2$ via intra- and inter-edge gap encodings, where the size of each gap is bounded by graph bandwidth $B$ (Chinn et al., 1982). We reduce this parameter via the bandwidth restriction scheme (Diamant et al., 2023). This prevents the model from learning dependencies among a vast vocabulary of size $N^2$. This improvement is more pronounced when compared with the existing edge list representations, as described in Figure 1c.

We present an autoregressive graph generative model to generate the proposed GEEL with node positional encoding. In detail, we observe that a simple LSTM (Hochreiter & Schmidhuber, 1997) combined with the proposed GEEL exhibits $O(M)$ complexity. Furthermore, combined with the node positional encoding that indicates the current node index, our GEEL achieved superior performance across ten general graph benchmarks while maintaining simplicity and scalability.

We further extend GEEL to attributed graphs by designing a new grammar and enforcing it to filter out invalid choices during generation. Specifically, our grammar specifies the position of node- and edge-types to be augmented in the GEEL representation. This approach led to competitive performance for two molecule generation benchmarks.

In summary, our key contributions are as follows:

- We newly introduce GEEL, a simple and scalable graph representation that has a compact representation size of $M$ based on edge lists while reducing the large vocabulary size $N^2$ of the edge lists to $B^2$ by applying gap encodings. We additionally reduce the graph bandwidth $B$ by the C-M ordering following Diamant et al. (2023).

- We propose to autoregressively generate GEEL by incorporating node positional encoding and combining it with an LSTM of $O(M)$ complexity.

- We extend GEEL to deal with attributed graphs by designing a new grammar that takes the node- and edge-types into account.

- We validate the efficacy and scalability of the proposed GEEL and the resulting generative framework by showing the state-of-the-art performance on twelve graph benchmarks.

## 2 RELATED WORK

**Adjacency matrix-based graph representation.** The adjacency matrix is the most prevalent graph representation, capturing straightforward pairwise relationships between nodes (Simonovsky & Komodakis, 2018; Jo et al., 2022; Vignac et al., 2022; You et al., 2018; Niu et al., 2020; Shi et al., 2020; Luo et al., 2021). For instance, You et al. (2018) proposed autoregressive generative models, Luo et al. (2021); Shi et al. (2020) presented normalizing flow models, and Jo et al. (2022) applied score-based models for graph generation. However, these methods suffer from the large representation size associated with generating the *full* adjacency matrix, which is impractical for large-scale graphs.

To solve this problem, several works have introduced scalable graph generative models (Liao et al., 2019; Dai et al., 2020; Diamant et al., 2023). Specifically, Liao et al. (2019) proposed a block-wise generation which enabled efficiency-quality trade-off. Dai et al. (2020) proposed to avoid consideration of every entry in the adjacency matrix, leveraging on the sparsity of graphs. Finally, Diamant et al. (2023) proposed to constrain the bandwidth via C-M ordering for any graph generative models, bypassing the out-of-bandwidth elements, which results in $NB$ representation size.

**Tree-based graph representation.** Researchers have developed tree-based representations by employing tree search algorithms (Segler et al., 2018; Ahn et al., 2022). Specifically, Segler et al. (2018) employed SMILES, a sequential representation for molecules, constructed from the DFS traversal of molecular graphs with omitted cycles. Complementing this, Ahn et al. (2022) designed a new representation that exploits the inherent tree-like structure of molecules.

**Motif-based graph representation.** Researchers have investigated motif-based representations (Jin et al., 2018; 2020; Guo et al., 2022), aiming to capture meaningful subgraphs with lower computational costs. In detail, Jin et al. (2018; 2020) focused on extracting common fragments from datasets. Since these techniques rely on domain-specific knowledge, Guo et al. (2022) introduced a domain-agnostic methodology to learn motif-based vocabulary by running reinforcement learning. However, it is still restricted to generating graphs with seen motifs that are included in the training set.

**Edge list-based graph representation.** A few works have presented edge list-based representations (Goyal et al., 2020; Bacciu & Podda, 2021). Employing an edge list as a graph representation reduces the representation size to $M$, which is smaller than that of the adjacency matrix, $N^2$. However, these methods suffer from the large vocabulary size $N^2$, resulting in a large search space and subsequently degrading the generation quality. They also face difficulties in capturing long-term dependencies due to their reliance on depth-first search (DFS) traversal for edge construction. Specifically, DFS traversal fails to closely place edges connected to the same node, so the model must span a broader range of steps to account for edges connected to the same node.

## 3 METHOD

In this section, we introduce our new graph representation, termed gap encoded edge list (GEEL), and autoregressive generation process with GEEL. GEEL has a small representation size $M$ by employing edge lists. In addition, GEEL enjoys a reduced vocabulary size with gap encodings and bandwidth restriction, narrowing down the search space and resulting in high-quality graph generation.

### 3.1 GAP ENCODED EDGE LIST REPRESENTATION (GEEL)

First, we present our GEEL representation, which leverages the small representation size of edge lists and the reduced vocabulary size with gap encoding and bandwidth restriction. Consider a graph with $N$ nodes, $M$ edges, and graph bandwidth $B$ (Chinn et al., 1982). The associated edge list has a representation size of $M$ which is smaller compared to the size of the adjacency matrix $N^2$. However, it has a large vocabulary size of $N^2$, consisting of tuples of node indices. To address this, we reduce the vocabulary size into $B^2$ by replacing the node indices in the original edge list with *gap encodings* as illustrated in Figure 1a. We encode two types of gaps: (1) the intra-edge gap between the source and the target nodes and (2) the inter-edge gap between source nodes in a pair of consecutive edges.

To this end, consider a connected undirected graph $G = (\mathcal{V}, \mathcal{E})$ with $N$ nodes and $M$ edges. We define the *ordering* as an invertible mapping $\pi : \mathcal{V} \rightarrow \{1, \ldots, N\}$ from a vertex into its rank for a

particular order of nodes. Then we define the *edge list* $\tau_{\text{EL}}$ as a sequence of pairs of integers:

$$\tau_{\text{EL}} = (s_1, t_1), (s_2, t_2), \ldots, (s_M, t_M),$$

where $s_m, t_m \in \{1, \ldots, N\}$ are the $m$-th source and target node indices that satisfy $(\pi^{-1}(s_m), \pi^{-1}(t_m)) \in \mathcal{E}$, respectively. Without loss of generality, we assume that $s_m < t_m$ and the edge list is sorted with respect to the ordering, i.e., if $m < \ell$, then $s_m < s_\ell$ or $s_m = s_\ell, t_m < t_\ell$. For example, $(1, 2), (1, 3), (2, 3), (3, 5)$ is a sorted edge list while $(1, 2), (2, 3), (3, 5), (1, 3)$ is not.

Consequently, we define our GEEL $\tau_{\text{GEEL}}$ as a sequence of gap encoding pairs as follows:

$$\tau_{\text{GEEL}} = (a_1, b_1), (a_2, b_2), \ldots, (a_M, b_M),$$

where $a_m$ and $b_m$ are the inter- and intra-edge gap encodings, respectively. To be specific, the inter-edge gap encoding indicates the difference between consecutive source indices as follows:

$$a_m = s_m - s_{m-1}, \qquad m = 1, \ldots, M, \qquad s_0 = 0.$$

Furthermore, the intra-edge gap encoding $b_m$ indicates the difference between the associated source and target node indices as follows:

$$b_m = t_m - s_m, \qquad m = 1, \ldots, M.$$

Then one can recover the original edge list $\tau_{\text{EL}}$ from GEEL $\tau_{\text{GEEL}}$ as follows:

$$s_m = \sum_{\ell=1}^{m} a_\ell, \qquad t_m = b_m + \sum_{\ell=1}^{m} a_\ell.$$

Note that the gap encodings are always positive and GEEL can be generalized to directed graphs by allowing negative intra-edge gap encodings.

**Reduction of the vocabulary size.** In training a generative model for edge lists and GEEL, the vocabulary size of $(s_m, t_m)$ and $(a_m, b_m)$ determines the complexity of the model. Here, we show that the vocabulary size of our GEEL is $B^2$ for the graph bandwidth $B$, which is smaller than the vocabulary size $N^2$ of the original edge list representation. Many real-world graphs, such as molecules and community graphs, exhibit low bandwidths as shown in Appendix C and by Diamant et al. (2023).

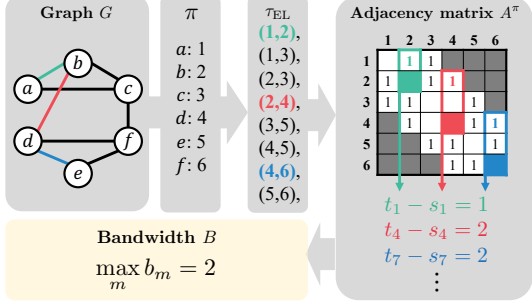

Figure 2: Bandwidth of an adjacency matrix.

The vocabulary size of our GEEL representation is bounded by $\max_m a_m \cdot \max_m b_m$. On one hand, the maximum intra-edge gap encoding coincides with the definition of the graph bandwidth, i.e., the maximum difference between a pair of adjacent nodes, denoted as $\max_m b_m = B$ as described in Figure 2. On the other hand, we can obtain the following upper bound for the inter-edge encoding:

$$\max_m a_m = \max_m (s_m - s_{m-1}) \leq \max_m \left( \max_{\ell < m} t_\ell - s_{m-1} \right) \leq \max_m \max_{\ell < m} (t_\ell - s_\ell) \leq B,$$

where the first inequality is based on deriving $s_m \leq \max_{\ell < m} t_\ell$ from the graph connectivity constraint: each source node index $s_m$ must appear as a target node index in prior for the graph to be connected, i.e., $s_m = t_\ell$ for some $\ell < m$. Consequently, the vocabulary size of our GEEL representation is upper-bounded by $\max_m a_m \cdot \max_m b_m \leq B^2$.

Given that the vocabulary size of GEEL is bounded by $B^2$, small bandwidth benefits graph generation by reducing the computational cost and the search space. We follow Diamant et al. (2023) to restrict the bandwidth via the Cuthill-McKee (C-M) node ordering (Cuthill & McKee, 1969). We also provide an ablation study with various node orderings in Section 4.3.

### 3.2 AUTOREGRESSIVE GENERATION OF GEEL AND NODE POSITIONAL ENCODING

**Autoregressive generation.** We first describe our method for the autoregressive generation of GEEL. To this end, we propose to maximize the evidence lower bound of the log-likelihood with respect to

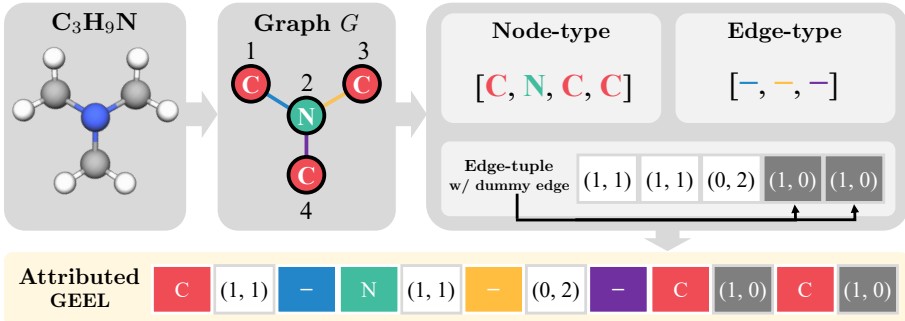

Figure 3: **An example of attributed GEEL.** The colored parts of the attributed GEEL denote the node features (i.e., C and N) and edge features (i.e., single bond -). The shaded parts denote the self-loops added to the original GEEL, where self-loops are added to the nodes that are not connected to the nodes with larger node indices (i.e., nodes with indices 3 and 4).

the latent ordering. To be specific, following prior works on autoregressive graph generative models (You et al., 2018; Liao et al., 2019; Dai et al., 2020), we maximize the following lower bound:

$$\log p(G) \geq \mathbb{E}_{q(\pi|G)}[\log p(G, \pi)] + C,$$

where $C$ is a constant and $q(\pi|G)$ is a variational posterior of the ordering given the graph $G$. Under this framework, our choice of choosing the C-M ordering for each graph corresponds to a choice of the variational distribution $q(\pi|G)$. Fixing a particular ordering for each graph yields the maximum log-likelihood objective for $\log p(G, \pi) = \log p(\tau_{\text{GEEL}})$.

We generate GEEL using an autoregressive model formulated as follows:

$$p(\tau_{\text{GEEL}}) = p(a_1, b_1) \prod_{m=2}^{M} p(a_m, b_m | \{a_\ell\}_{\ell=1}^{m-1}, \{b_\ell\}_{\ell=1}^{m-1}).$$

Notably, we treat each tuple $(a_m, b_m)$ as one token and generate a token at each step. Similar to text generative models, we also introduce the begin-of-sequence (BOS) and the end-of-sequence (EOS) tokens to indicate the start and end of the sequence generation process, respectively (Collins, 2003).

Finally, it is noteworthy that we train a long short-term memory (LSTM) model (Hochreiter & Schmidhuber, 1997) to minimize the proposed objective. Adopting LSTM as our backbone ensures an $O(M)$ complexity for our generative model, due to the linear complexity of the LSTM. The model architecture can be freely changed to other architectures as demonstrated in Section 4.3.

**Source node positional encoding.** While the gap encoding allows a significant reduction in vocabulary size, it also complicates the inherent semantics since each source node index is represented by the cumulative summation over the intra-edge gap encodings. Instead of burdening the generative model to learn the cumulative summation, we directly supplement the token embeddings with the node positional encoding of the source node index, i.e., $\sum_{\ell=1}^{m} a_\ell$ at the $(m+1)$-th step as:

$$\phi\big((a_m, b_m)\big) = \phi_{\text{tuple}}\big((a_m, b_m)\big) + \phi_{\text{PE}}\Big(\sum_{\ell=1}^{m} a_\ell\Big),$$

where $\phi$ is the final embedding, $\phi_{\text{tuple}}$ is the token embedding, and $\phi_{\text{PE}}$ is the positional encoding.

### 3.3 GEEL FOR ATTRIBUTED GRAPHS

In this section, we elaborate on the extension of GEEL to attributed graphs. To this end, we augment the GEEL representation with node- and edge-types. Our attributed GEEL follows a specific grammar that filters out invalid choices of tokens.

**Grammar of attributed GEEL.** For the generation of attributed graphs with node- and edge-types, we not only generate the edge-tuples $(a_k, b_k)$ as in Section 3.1 but also generate node- and edge-types according to the following rules. We provide an illustrative example of attribute GEEL in Figure 3.

- Before describing edge-tuples starting with a new source node, add the paired node-type.
- After adding an edge-tuple, add the paired edge-type.

One can observe that our rules are intuitive: for each source node, we first describe the node-type and then generate the associated edge-tuple and types. For nodes that are not associated with any edge-tuple as a source node, we add a "dummy" edge-tuple with the node as its source. As a result, our representation size for attributed graphs is at most $2M + N$ and the vocabulary size is $2B + |X| + |E|$ where $|X|$ denotes the number of node features and $|E|$ denotes the number of edge features.

**Autoregressive generation with grammar constraints.** To enforce the attribute grammar, we introduce an algorithm to filter out invalid choices of tokens.

- The first token is always the node-type token.
- The node-type tokens are always followed by edge-tuple tokens.
- The edge-tuple tokens are always followed by edge-type tokens.
- The edge-type tokens are always followed by node-type tokens or edge-tuple tokens.

These rules prevent the generation process from generating invalid GEEL such as the list that consists of only node-types or the list that has an edge-tuple without a following edge-type token. This procedure is done by computing the probability only over valid choices.

## 4 EXPERIMENT

### 4.1 GENERAL GRAPH GENERATION

**Evaluation protocol.** We adopt maximum mean discrepancy (MMD) as our evaluation metric to compare three graph property distributions between test graphs and the same number of generated graphs: degree, clustering coefficient, and 4-node-orbit counts. **Results that are either superior to or comparable with the MMD of training graphs** are highlighted in bold in Table 1. The comparability of MMD values is determined by examining whether the MMD falls within a range of one standard deviation. Notably, our work stands out as a baseline for graph generative models, given its comprehensive evaluation across ten diverse graph datasets and its state-of-the-art performance. Further details regarding our experimental setup are in Appendix A.

We validate the general graph generation performance of our GEEL on eight general graph datasets with varying sizes. Four small-sized graphs are: (1) **Planar**, 200 planar graphs, (2) **Lobster**, 100 Lobster graphs (Senger, 1997), (3) **Enzymes** (Schomburg et al., 2004), 587 protein tertiary structure graphs, and (4) **SBM**, 200 stochastic block model graphs. Four large-sized graphs are: (5) **Ego**, 757 large Citeseer network dataset (Sen et al., 2008), (6) **Grid**, 100 2D grid graphs, (7) **Proteins**, 918 protein graphs, and (8) **3D point cloud**, 41 3D point cloud graphs of household objects. Additional results on two smaller datasets (**Ego-small** and **Community-small**) are provided in Appendix E.

We compare our GEEL with seventeen deep graph generative models. They can be categorized into two according to the type of representation they use. On one hand, fifteen adjacency matrix-based methods are: GraphVAE (Simonovsky & Komodakis, 2018), GraphRNN (You et al., 2018), GNF Liu et al. (2019), GRAN (Liao et al., 2019), EDP-GNN (Niu et al., 2020), GraphAF (Shi et al., 2020), GraphDF (Luo et al., 2021), SPECTRE (Martinkus et al., 2022), BiGG (Dai et al., 2020), GDSS (Jo et al., 2022), DiGress (Vignac et al., 2022), GDSM (Luo et al., 2022), GraphARM (Kong et al., 2023), BwR (Diamant et al., 2023), and SwinGNN (Yan et al., 2023). On the other hand, two edge list-based methods are GraphGen (Goyal et al., 2020) and GraphGen-Redux (Bacciu & Podda, 2021). Note that one graph compression-based method HGGT (Jang et al., 2024) is also included. We provide a detailed implementation description in Appendix B.

**Generation quality.** We provide experimental results in Table 1. We observe that the proposed GEEL consistently shows superior or competitive results across all the datasets. This verifies the ability of our model to effectively capture the topological information of both large and small graphs. The visualization of generated samples can be found in Appendix D. It is worth noting that the generation performance on small graphs has reached a saturation point, yielding results that are either superior or comparable to training graphs.

Table 1: **General graph generation performance.** The baseline results are from prior works (Jo et al., 2022; Kong et al., 2023; Martinkus et al., 2022; Dai et al., 2020; Diamant et al., 2023) or obtained by running the open-source codes. Note that OOM indicates Out-Of-Memory and N.A. indicates that the generated samples are all invalid. For each metric, the numbers that are superior or comparable to the MMD of the training graphs are highlighted in **bold**. The comparability is determined by whether the MMD falls within one standard deviation.

| Method | Planar $\|V\|=64$ | | | Lobster $10 \le \|V\| \le 100$ | | | Enzymes $10 \le \|V\| \le 125$ | | | SBM $31 \le \|V\| \le 187$ | | |
|---|---|---|---|---|---|---|---|---|---|---|---|---|
| | Deg. | Clus. | Orb. | Deg. | Clus. | Orb. | Deg. | Clus. | Orb. | Deg. | Clus. | Orb. |
| Training | 0.001 | 0.002 | 0.000 | 0.005 | 0.000 | 0.007 | 0.006 | 0.018 | 0.007 | 0.016 | 0.002 | 0.047 |
| GraphVAE | - | - | - | - | - | - | 1.369 | 0.629 | 0.191 | - | - | - |
| GraphRNN | 0.005 | 0.278 | 1.254 | **0.000** | **0.000** | **0.000** | 0.017 | 0.062 | 0.046 | **0.006** | 0.058 | 0.079 |
| GRAN | **0.001** | 0.043 | **0.001** | 0.038 | 0.000 | 0.001 | 0.023 | 0.031 | 0.169 | **0.011** | 0.055 | **0.054** |
| EDP-GNN | - | - | - | - | - | - | 0.023 | 0.268 | 0.082 | - | - | - |
| GraphGen | 1.762 | 1.423 | 1.640 | 0.548 | 0.040 | 0.247 | 0.146 | 0.079 | 0.054 | 1.230 | 1.752 | 0.597 |
| GraphGen-Redux | 1.105 | 1.809 | 0.517 | 1.189 | 1.859 | 0.885 | 0.456 | 0.035 | 0.251 | - | - | - |
| GraphAF | - | - | - | - | - | - | 1.669 | 1.283 | 0.266 | - | - | - |
| GraphDF | - | - | - | - | - | - | 1.503 | 1.283 | 0.266 | - | - | - |
| BiGG | 0.002 | 0.004 | **0.000** | **0.000** | **0.000** | **0.000** | 0.010 | **0.018** | 0.011 | 0.029 | **0.003** | **0.036** |
| GDSS | 0.250 | 0.393 | 0.587 | 0.117 | 0.002 | 0.149 | 0.026 | 0.061 | **0.009** | 0.496 | 0.456 | 0.717 |
| DiGress | **0.000** | **0.002** | 0.008 | **0.021** | **0.000** | **0.004** | 0.011 | 0.039 | 0.010 | **0.006** | 0.051 | **0.058** |
| GDSM | - | - | - | - | - | - | 0.013 | 0.088 | 0.010 | - | - | - |
| GraphARM | - | - | - | - | - | - | 0.029 | 0.054 | 0.015 | - | - | - |
| BwR | 0.609 | 0.542 | 0.097 | 0.316 | **0.000** | 0.247 | 0.021 | 0.095 | 0.025 | | N.A. | |
| HGGT | **0.000** | **0.001** | **0.000** | 0.003 | **0.000** | 0.015 | **0.005** | **0.017** | **0.000** | 0.017 | 0.014 | 0.076 |
| GEEL (ours) | **0.001** | 0.010 | **0.001** | **0.002** | **0.000** | **0.001** | **0.005** | **0.018** | **0.006** | 0.025 | **0.003** | **0.026** |

(a) Small graphs ($\|V\|_{\max} \le 187$)

| Method | Ego $50 \le \|V\| \le 399$ | | | Grid $100 \le \|V\| \le 400$ | | | Proteins $13 \le \|V\| \le 1575$ | | | 3D point cloud $8 \le \|V\| \le 5037$ | | |
|---|---|---|---|---|---|---|---|---|---|---|---|---|
| | Deg. | Clus. | Orb. | Deg. | Clus. | Orb. | Deg. | Clus. | Orb. | Deg. | Clus. | Orb. |
| Training | 0.010 | 0.003 | 0.016 | 0.000 | 0.000 | 0.000 | 0.002 | 0.003 | 0.002 | 0.004 | 0.090 | 0.015 |
| GraphVAE | - | - | - | 1.619 | **0.000** | 0.919 | - | - | - | | OOM | |
| GraphRNN | 0.117 | 0.615 | 0.043 | 0.011 | 0.000 | 0.021 | 0.011 | 0.140 | 0.880 | | OOM | |
| GRAN | 0.026 | 0.342 | 0.254 | 0.001 | 0.004 | 0.002 | 0.002 | 0.049 | 0.130 | 0.018 | 0.510 | 0.210 |
| EDP-GNN | - | - | - | 0.455 | 0.238 | 0.328 | - | - | - | - | - | - |
| GraphGen | 0.578 | 1.199 | 0.776 | 1.550 | 0.017 | 0.860 | 1.392 | 1.743 | 0.866 | | OOM | |
| GraphGen-Redux | 1.088 | 0.702 | 0.155 | - | - | - | - | - | - | - | - | - |
| SPECTRE | - | - | - | - | - | - | 0.013 | 0.047 | 0.029 | - | - | - |
| BiGG | **0.010** | 0.017 | **0.012** | **0.000** | **0.000** | 0.001 | **0.001** | 0.026 | 0.023 | **0.003** | 0.210 | **0.007** |
| GDSS | 0.393 | 0.873 | 0.209 | 0.111 | 0.005 | 0.070 | 0.703 | 1.444 | 0.410 | | OOM | |
| DiGress | 0.063 | 0.031 | 0.024 | 0.016 | **0.000** | 0.004 | | OOM | | | OOM | |
| GDSM | - | - | - | 0.002 | 0.000 | 0.000 | - | - | - | - | - | - |
| BwR | | N.A. | | 0.385 | 1.187 | 0.083 | 0.092 | 0.229 | 0.489 | 1.820 | 1.295 | 0.869 |
| SwinGNN | - | - | - | **0.000** | **0.000** | **0.000** | **0.002** | 0.016 | **0.003** | - | - | - |
| HGGT | | N.A. | | **0.000** | **0.000** | **0.000** | 0.098 | 0.708 | 0.619 | | OOM | |
| GEEL (ours) | 0.053 | 0.017 | **0.016** | **0.000** | **0.000** | **0.000** | **0.003** | **0.005** | **0.003** | 0.002 | 0.081 | 0.020 |

(b) Large graphs ($399 \le \|V\|_{\max} \le 5037$)

**Scalability analysis.** Next, we empirically validate the time complexity of our model. We first verify if the actual inference time aligns well with the theoretical $O(M)$ curve. To this end, we generated Grid graphs with varying numbers of nodes: [10, 100, 200, 500, 1k, 2k, 5k, 10k]. The results shown in Figure 4 indicate an alignment between the actual inference time and the ideal curve.

Then we conduct further experiments to compare the inference time of our model with that of other baselines. Note that we used the same computational resource for all models and other experimental details are in Appendix B. The results presented in Table 2 represent the time required to generate a single sample. Notably, our model shows a shorter inference time owing to the compactness of our representation, GEEL, even compared to other scalable graph generative models (Liao et al., 2019; Dai et al., 2020). This evidence underscores the scalability advantages of our GEEL.

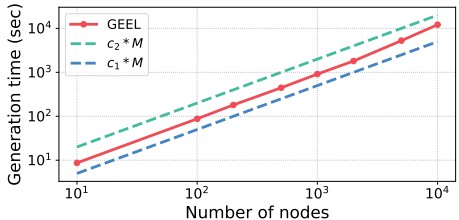

Figure 4: **Infer. time on various graph sizes.**

Table 2: **Inference time (sec) to generate one graph**.

| Method | Type | Enzymes | Planar | Grid |
|---|---|---|---|---|
| GRAN | Auto. Reg. | 0.24 | 0.66 | 1.39 |
| BiGG | Auto. Reg. | 2.70 | 0.61 | 3.58 |
| GDSS | Diffusion | 15.21 | 15.25 | 30.89 |
| DiGress | Diffusion | 8.82 | 12.09 | 32.29 |
| GEEL (ours) | Auto. Reg. | **0.12** | **0.25** | **1.11** |

Table 4: **Molecular graph generation performance of the QM9 and ZINC datasets.** The baseline results are from prior works (Jo et al., 2022; Ahn et al., 2022). The best results of molecule-specific generative models and domain-agnostic generative models are both highlighted in **bold**.

| | QM9 | | | | | | ZINC250k | | | | | |
|---|---|---|---|---|---|---|---|---|---|---|---|---|
| Method | Val. (%) ($\uparrow$) | NSPDK ($\downarrow$) | FCD ($\downarrow$) | Scaf. ($\uparrow$) | SNN ($\uparrow$) | Frag. ($\uparrow$) | Val. (%) ($\uparrow$) | NSPDK ($\downarrow$) | FCD ($\downarrow$) | Scaf. ($\uparrow$) | SNN ($\uparrow$) | Frag. ($\uparrow$) |
| | | | | | Molecule-specific generative models | | | | | | | |
| CharRNN | 99.57 | **0.0003** | **0.087** | 0.9313 | 0.5162 | 0.9887 | 96.95 | **0.0003** | 0.474 | 0.4024 | 0.3965 | **0.9988** |
| CG-VAE | **100.0** | - | 1.852 | 0.6628 | 0.3940 | 0.9484 | **100.0** | - | 11.335 | 0.2411 | 0.2656 | 0.8118 |
| MoFlow | 91.36 | 0.0169 | 4.467 | 0.1447 | 0.3152 | 0.6991 | 63.11 | 0.0455 | 20.931 | 0.0133 | 0.2352 | 0.7508 |
| STGG | **100.0** | - | 0.585 | 0.9416 | **0.9998** | 0.9984 | **100.0** | - | **0.278** | **0.7192** | **0.4664** | 0.9932 |
| | | | | | Domain-agnostic graph generative models | | | | | | | |
| EDP-GNN | 47.52 | 0.0046 | 2.680 | 0.3270 | 0.5265 | 0.8313 | 63.11 | 0.0485 | 16.737 | 0.0000 | 0.0815 | 0.0000 |
| GraphAF | 74.43 | 0.0207 | 5.625 | 0.3046 | 0.4040 | 0.8319 | 68.47 | 0.0442 | 16.023 | 0.0672 | 0.2422 | 0.5348 |
| GraphDF | 93.88 | 0.0636 | 10.928 | 0.0978 | 0.2948 | 0.4370 | 90.61 | 0.1770 | 33.546 | 0.0000 | 0.1722 | 0.2049 |
| GDSS | 95.72 | 0.0033 | 2.900 | 0.6983 | 0.3951 | 0.9224 | 97.01 | 0.0195 | 14.656 | 0.0467 | 0.2789 | 0.8138 |
| DiGress | 98.19 | 0.0003 | 0.095 | 0.9353 | 0.5263 | 0.0023 | 94.99 | 0.0021 | 3.482 | 0.4163 | 0.3457 | 0.9679 |
| DruM | 99.69 | **0.0002** | 0.108 | **0.9449** | **0.5272** | 0.9867 | 98.65 | **0.0015** | 2.257 | 0.5299 | 0.3650 | 0.9777 |
| GraphARM | 90.25 | 0.0020 | 1.220 | - | - | - | 88.23 | 0.0550 | 16.260 | - | - | - |
| GEEL (ours) | **100.0** | **0.0002** | **0.089** | 0.9386 | 0.5161 | **0.9891** | **99.31** | 0.0068 | **0.401** | **0.5565** | **0.4473** | **0.992** |

In addition, we provide the reduced representation and vocabulary sizes in Table 3. Note that the vocabulary size of the original edge list and the representation size of the adjacency matrix are both $N^2$. We can observe that GEEL is efficient in both sizes.

### 4.2 MOLECULAR GRAPH GENERATION

**Experimental setup.** We use two molecular datasets: QM9 (Ramakrishnan et al., 2014) and ZINC250k (Irwin et al., 2012). Following the previous work (Jo et al., 2022), we evaluate 10,000 generated molecules using six metrics: (a) the ratio of valid molecules without correc-

Table 3: **Vocabulary and representation sizes**. The vocabulary size is $B^2$ and the representation size is $M$ where $B$ is bandwidth, $N$ is the number of nodes, and $M$ is the number of edges.

| Dataset | Vocab. size | Rep. size | $N^2$ |
|---|---|---|---|
| Planar | 676 | 181 | 4096 |
| Lobster | 2401 | 99 | 9604 |
| Enzymes | 361 | 149 | 15625 |
| SBM | 12321 | 1129 | 34969 |
| Ego | 58081 | 1071 | $> 10^6$ |
| Grid | 361 | 684 | 467856 |
| Proteins | 62500 | 1575 | $> 10^6$ |
| 3D point cloud | 111556 | 10886 | $> 10^7$ |

tion (**Val.**), (b) neighborhood subgraph pairwise distance kernel (**NSPDK**), (c) Frechet ChemNet Distance (**FCD**) (Preuer et al., 2018), (d) scaffold similarity (**Scaf.**), (e) similarity to the nearest neighbor (**SNN**), and (f) fragment similarity (**Frag.**). We use the same split with Jo et al. (2022) for a fair comparison. Note that in contrast to other general graph generation methods, our approach uniquely facilitates the direct representation of ions by employing them as a node type. We provide details in Appendix A.

**Baselines.** We compare GEEL with seven general deep graph generative models: EDP-GNN (Niu et al., 2020), GraphAF (Shi et al., 2020), GraphDF (Luo et al., 2021), GDSS (Jo et al., 2022), DiGress (Vignac et al., 2022), DruM (Jo et al., 2023), and GraphARM (Kong et al., 2023). In addition, for further comparison, we also compare GEEL with four molecule-specific generative models: CharRNN (Segler et al., 2018), CG-VAE (Jin et al., 2020), MoFlow (Zang & Wang, 2020), and STGG (Ahn et al., 2022). We provide a detailed implementation description in Appendix B.

**Results.** The experimental results are reported in Table 4. We observe that our GEEL shows superior results to domain-agnostic graph generative models and competitive results with molecule-specific generative models. In particular, for the QM9 dataset, we observe that our GEEL shows superior

Table 5: **Average MMD results for different model architectures.**

| Backbone | Planar | Enzymes | Grid |
|---|---|---|---|
| LSTM | 0.002 | 0.009 | 0.000 |
| Transformer | 0.003 | 0.008 | 0.000 |

Table 6: **Average MMD for different representations.** We adopted LSTM as a model architecture and OOM denotes out-of-memory error.

| Representation | Repr. | Vocab. | Com.-small | Grid | Point |
|---|---|---|---|---|---|
| Flattened adj. | $N^2$ | 2 | 0.029 | OOM | OOM |
| Edge list | $M$ | $N^2$ | 0.010 | 0.000 | OOM |
| Edge list + intra gap | $M$ | $NB$ | 0.013 | 0.000 | OOM |
| GEEL (ours) | $M$ | $B^2$ | 0.016 | 0.000 | 0.044 |

results on FCD and Scaffold scores even compared to the molecule-specific models. We also provide the visualization of generated molecules in Appendix D.

### 4.3 ABLATION STUDIES

**Different model architectures.** Here, we discuss the results of generating GEEL with Transformers (Vaswani et al., 2017). We evaluate four datasets: Planar, Lobster, Enzymes, and Grid, employing three MMD metrics for assessment. As presented in Table 5, LSTM shows competitive results to Transformers. Notably, LSTM achieves this performance with significantly reduced computational cost, having a linear complexity of $O(n)$, in contrast to the quadratic complexity $O(n^2)$ of Transformers, where $n$ represents the sequence length.

**Different representations.** We discuss the results of generating graphs with various representations here. We compare our GEEL with three alternative representations: flattened adjacency matrix, the edge list, and the edge list with intra-edge gap. The edge list is the edge list with traditional edge encoding (without any gap encoding) and the last one is an edge list wherein the target node of each edge is substituted by its intra-edge gap. Note that the edge lists are sorted in the same way we sort the edge list, as explained in Section 3.1. The comparative results are in Table 6. We can observe that GEEL effectively reduces the vocabulary size compared to other representations. This enables the generation of large-scale graphs, such as 3D point clouds, without memory constraints.

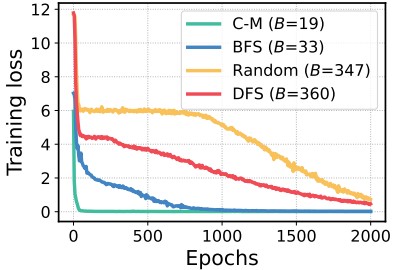

Figure 5: **Training curve with various node orderings.**

**Different node orderings.** We here assess the effect of node ordering on graph generation. We compare our C-M ordering to BFS, DFS, and random ordering using the Grid dataset. As illustrated in Figure 5, the C-M ordering outperforms others with faster convergence of training loss. Notably, the BFS also shows competitive loss convergence with C-M as it mitigates the burden of long-term dependency. Specifically, both C-M and BFS orderings position edges related to the same node more closely than other baselines. These results highlight the effectiveness of C-M ordering on bandwidth reduction and generating high-quality graphs. Note that the MMD performance with any ordering eventually converges to the same levels of MMD as explained in Appendix H.

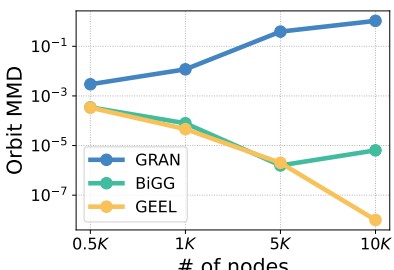

Figure 6: **Orbit MMD with various graph sizes.**

**Quality with various graph sizes.** We also evaluate the generated graph quality with respect to the graph size. Following a prior work (Dai et al., 2020), we conduct experiments on grid data with {0.5k, 1k, 5k, 10k} nodes and reported orbit MMD. The results are in Figure 6 and we can see GEEL preserves high quality on large-scale graphs with up to 10k nodes.

## 5 CONCLUSION

In this work, we introduce GEEL, an edge list-based graph representation that is both simple and scalable. By combining GEEL with an LSTM, our graph generative model achieves an $O(M)$ complexity, showing a significant enhancement in generation quality and scalability over prior graph generative models.

**Reproducibility** All experimental code related to this paper is available at https://github.com/yunhuijang/GEEL. Detailed insights regarding the experiments, encompassing dataset and model specifics, are available in Section 4. For intricate details like hyperparameter search, consult Appendix A. In addition, the reproduced dataset for each baseline is in Appendix B.

**Acknowledgements** This work partly was supported by Institute of Information & communications Technology Planning & Evaluation (IITP) grant funded by the Korea government (MSIT) (No. IITP-2019-0-01906, Artificial Intelligence Graduate School Program (POSTECH)), the National Research Foundation of Korea (NRF) grant funded by the Korea government (MSIT) (No. 2022R1C1C1013366), and Basic Science Research Program through the National Research Foundation of Korea (NRF) funded by the Ministry of Education (2022R1A6A1A0305295413).

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

# A    EXPERIMENTAL DETAILS

In this section, we provide the details of the experiments.

## A.1    GENERAL GRAPH GENERATION

Table 7: **Hyperparameters of GEEL in general graph generation.**

|               | Planar | Lobster | Enzymes | SBM    | Ego    | Grid  | Proteins | 3D point cloud |
|---------------|--------|---------|---------|--------|--------|-------|----------|----------------|
| Learning rate | 0.001  | 0.0001  | 0.0005  | 0.0001 | 0.0001 | 0.001 | 0.0001   | 0.0012         |
| Batch size    | 128    | 64      | 128     | 8      | 8      | 64    | 8        | 4              |

Table 8: **Default hyperparameters of GEEL.**

| Input dropout rate | Dim. of token embedding | Num. of layers |
|--------------------|-------------------------|----------------|
| 0.1                | 512                     | 3              |

We used the same split with GDSS (Jo et al., 2022) for Enzymes and Grid datasets, with DiGress (Vignac et al., 2022) for Planar and SBM datasets, with BiGG (Dai et al., 2020) for Lobster, Proteins, and 3D point cloud datasets, and with GraphRNN (You et al., 2018) for ego dataset. We perform the hyperparameter search to choose the best learning rate in {0.0001, 0.0005, 0.001, and 0.0012}. We select the model with the best MMD with the lowest average of three graph statistics: degree, clustering coefficient, and 4-orbit count. In addition, we report the means of 5 different runs. It is notable that GEEL samples a C-M ordering for a graph at each training step (instead of fixing a unique ordering per graph). We found this to improve novelty without any decrease in performance and changing the hyper-parameters. We provide the best learning rates in Table 7 and other default hyperparameters that we have not tuned in Table 8.

## A.2    MOLECULAR GRAPH GENERATION

We used the same split with GDSS (Jo et al., 2022) for a fair evaluation. Following general graph generation, we perform the hyperparameter search to choose the best learning rate in {0.0001, 0.001} and select the model with the best FCD score. The best learning rates are 0.001 for both QM9 and ZINC datasets and other default hyperparameters are in Table 8 which is the same as the general graph generation.

For ion tokenization, we used 13 node tokens for QM9: [C-], [CH-], [C], [F], [N+], [N-], [NH+], [NH2+], [NH3+], [NH], [N], [O-], [O]. In addition, we used 29 tokens for ZINC: [Br], [CH-], [CH2-], [CH], [C], [Cl], [F], [I], [N+], [N-], [NH+], [NH-], [NH2+], [NH3+], [NH], [N], [O+], [O-], [OH+], [O], [P+], [PH+], [PH2], [PH], [P], [S+], [S-], [SH+], [S].

# B IMPLEMENTATION DETAILS

## B.1 COMPUTING RESOURCES

We used Pytorch (Paszke et al., 2019) to implement GEEL and trained the LSTM (Hochreiter & Schmidhuber, 1997) models on GeForce RTX 3090 GPU. Note that we used A100-40GB for the 3D point cloud dataset. In addition, due to the CUDA compatibility issue of BiGG (Dai et al., 2020), we used GeForce GTX 1080 Ti GPU and 40 CPU cores for all models for inference time evaluation in Figure 1b and Table 2.

## B.2 DETAILS FOR BASELINE IMPLEMENTATION

Table 9: **Reproduced dataset for each baseline.**

|          | Planar | Lobster | Enzymes | SBM | Ego | Grid | Proteins | 3D point cloud |
|----------|--------|---------|---------|-----|-----|------|----------|----------------|
| GRAN     | -      | -       | ✓       | -   | ✓   | -    | -        | -              |
| GraphGen | ✓      | ✓       | ✓       | ✓   | ✓   | ✓    | ✓        | ✓              |
| BiGG     | ✓      | -       | ✓       | ✓   | ✓   | -    | -        | -              |
| GDSS     | ✓      | ✓       | -       | ✓   | ✓   | -    | ✓        | ✓              |
| DiGress  | -      | ✓       | ✓       | -   | ✓   | ✓    | ✓        | ✓              |
| BwR      | ✓      | ✓       | ✓       | ✓   | ✓   | ✓    | ✓        | ✓              |

**General graph generation.** The baseline results from prior works are as follows. We reproduced GRAN (Liao et al., 2019), GraphGen (Goyal et al., 2020), DiGress (Vignac et al., 2022), GDSS (Jo et al., 2022), BiGG (Dai et al., 2020), and BwR (Diamant et al., 2023) for the datasets that are not reported in the original paper using their open-source codes. The reproduced datasets are explained in Table 9. Note that BwR results are based on the GraphRNN variant, which shows the best results through three provided baselines. The other results for GraphVAE (Simonovsky & Komodakis, 2018), GNF (Liu et al., 2019), EDP-GNN (Niu et al., 2020), GraphAF (Shi et al., 2020), GraphDF (Luo et al., 2021), and GDSS (Jo et al., 2022) are obtained from GDSS, while the results for GRAN (Liao et al., 2019), GraphRNN (You et al., 2018), and BiGG (Dai et al., 2020) are from BiGG and SPECTRE (Martinkus et al., 2022). Finally, the remaining results for SPECTRE and GDSM (Luo et al., 2022) are derived from their respective paper. We used original hyperparameters when the original work provided them.

**Molecular graph generation.** The baseline results from prior works are as follows. The results for five domain-agnostic graph generative models: EDP-GNN (Niu et al., 2020), GraphAF (Shi et al., 2020), GraphDF (Luo et al., 2021), GDSS (Jo et al., 2022), DruM (Jo et al., 2023) are from DruM, and the GraphARM (Kong et al., 2023) result is extracted from the corresponding paper. Moreover, we reproduced DiGress (Vignac et al., 2022) using their open-source codes.

In addition, for molecule generative models, the result of MoFlow (Zang & Wang, 2020) is from DruM, the results of CG-VAE (Jin et al., 2020) and STGG (Ahn et al., 2022) is from STGG. Furthermore, we reproduced CharRNN (Segler et al., 2018).

## B.3 DETAILS FOR INFERENCE TIME ANALYSIS

We conducted the inference time analysis using the same GeForce GTX 1080 Ti GPU for all models. The batch size is 10 and the inference time to generate a single graph is computed by dividing the total inference time by the batch size.

## B.4 DETAILS FOR THE IMPLEMENTATION

We adapted node ordering code from (Diamant et al., 2023), evaluation scheme from (Jo et al., 2022; Martinkus et al., 2022), and NSPDK computation from (Goyal et al., 2020).

## C  GRAPH STATISTICS OF DATASETS

### C.1  GENERAL GRAPHS

Table 10: **Statstics of general datasets.**

| Dataset | # of graphs | # of nodes | Max. $B$ | Max. # of edges |
|---|---|---|---|---|
| Planar | 200 | $\lvert V \rvert = 64$ | 26 | 181 |
| Lobster | 100 | $10 \leq \lvert V \rvert \leq 100$ | 49 | 99 |
| Enzymes | 587 | $10 \leq \lvert V \rvert \leq 125$ | 19 | 149 |
| SBM | 200 | $31 \leq \lvert V \rvert \leq 187$ | 111 | 1129 |
| Ego | 757 | $50 \leq \lvert V \rvert \leq 399$ | 241 | 1071 |
| Grid | 100 | $100 \leq \lvert V \rvert \leq 400$ | 19 | 684 |
| Proteins | 918 | $13 \leq \lvert V \rvert \leq 1575$ | 125 | 1575 |
| 3D point cloud | 41 | $8 \leq \lvert V \rvert \leq 5037$ | 167 | 10886 |

Table 11: **Standard deviation of MMD in training dataset.**

| Planar | | | Lobster | | | Enzymes | | | SBM | | |
|---|---|---|---|---|---|---|---|---|---|---|---|
| Deg. | Clus. | Orb. | Deg. | Clus. | Orb. | Deg. | Clus. | Orb. | Deg. | Clus. | Orb. |
| 0.000 | 0.001 | 0.000 | 0.003 | 0.000 | 0.006 | 0.001 | 0.003 | 0.002 | 0.008 | 0.002 | 0.017 |

(a) Small graphs ($\lvert V \rvert_{\max} \leq 187$)

| Ego | | | Grid | | | Proteins | | | 3D point cloud | | |
|---|---|---|---|---|---|---|---|---|---|---|---|
| Deg. | Clus. | Orb. | Deg. | Clus. | Orb. | Deg. | Clus. | Orb. | Deg. | Clus. | Orb. |
| 0.005 | 0.001 | 0.004 | 0.000 | 0.000 | 0.000 | 0.001 | 0.002 | 0.001 | 0.04 | 0.062 | 0.017 |

(b) Large graphs ($399 \leq \lvert V \rvert_{\max} \leq 5037$)

The statistics of general graphs are summarized in Table 10. It is notable that the bandwidths are relatively low compared to the number of nodes for real-world graphs, which enables the reduction of the vocabulary size of GEEL. In addition, we provide the standard deviations of MMD of training graphs that we used as a criterion to verify comparability in Table 11.

### C.2  MOLECULAR GRAPHS

Table 12: **Statstics of molecular datasets: QM9 and ZINC250k.**

| Dataset | # of graphs | # of nodes | Max. $B$ | Max. # of edges | # of node types | # of edge types |
|---|---|---|---|---|---|---|
| QM9 | 133,885 | $1 \leq \lvert V \rvert \leq 9$ | 5 | 13 | 13 | 4 |
| ZINC250k | 249,455 | $6 \leq \lvert V \rvert \leq 38$ | 10 | 45 | 29 | 4 |

The statistics of molecular graphs are summarized in Table 12. Note that the # of node types indicate the number of ionized node type tokens as explained in Appendix A.

# D  GENERATED SAMPLES

## D.1  GENERAL GRAPH GENERATION

### Planar

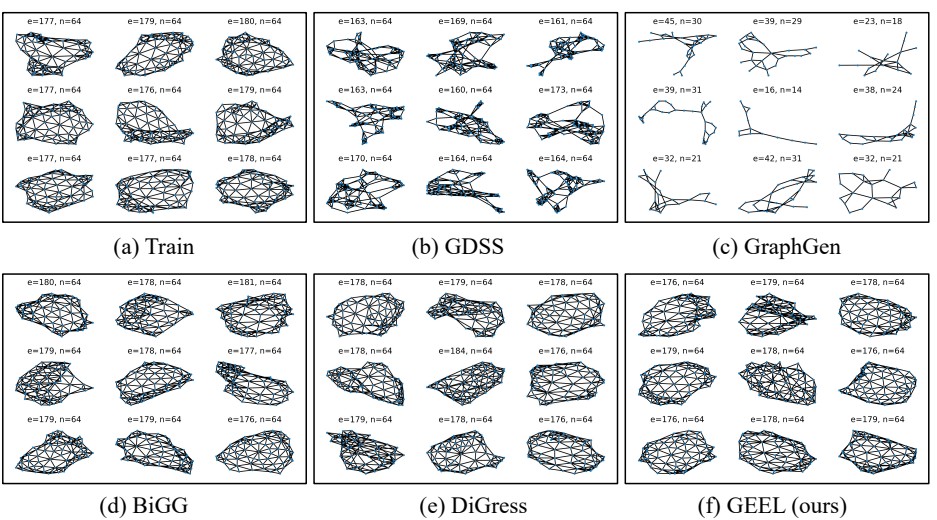

Figure 7: **Visualization of the graphs from the Planar dataset and the generated graphs.**

### Lobster

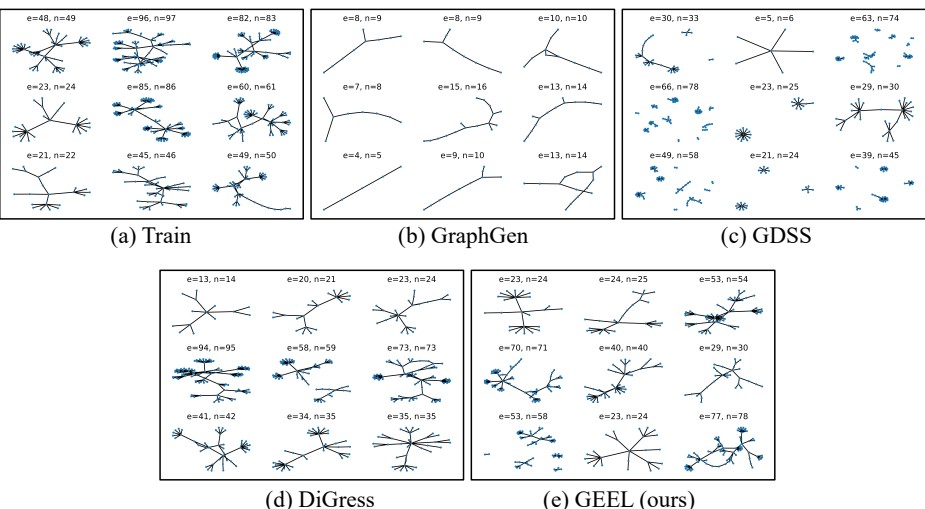

Figure 8: **Visualization of the graphs from the Lobster dataset and the generated graphs.**

## Enzymes

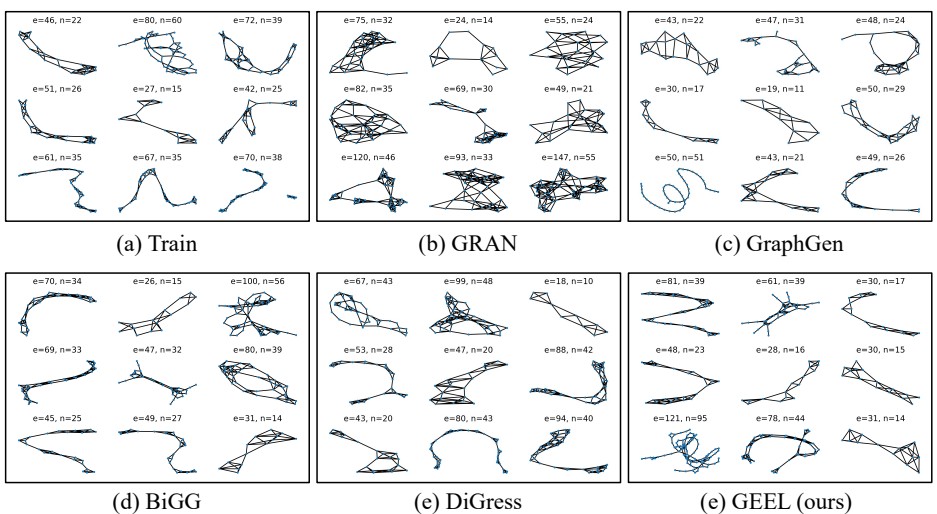

Figure 9: **Visualization of the graphs from the Enzymes dataset and the generated graphs.**

## SBM

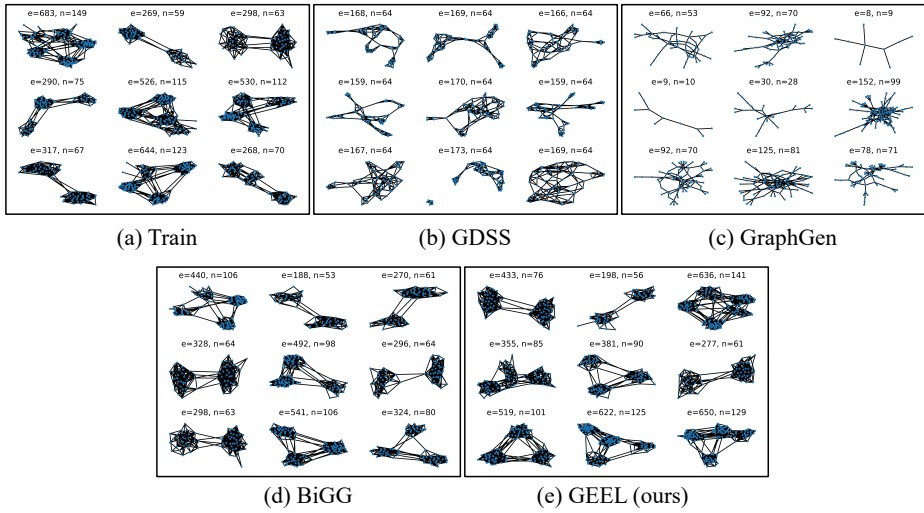

Figure 10: **Visualization of the graphs from the SBM dataset and the generated graphs.**

**Ego**

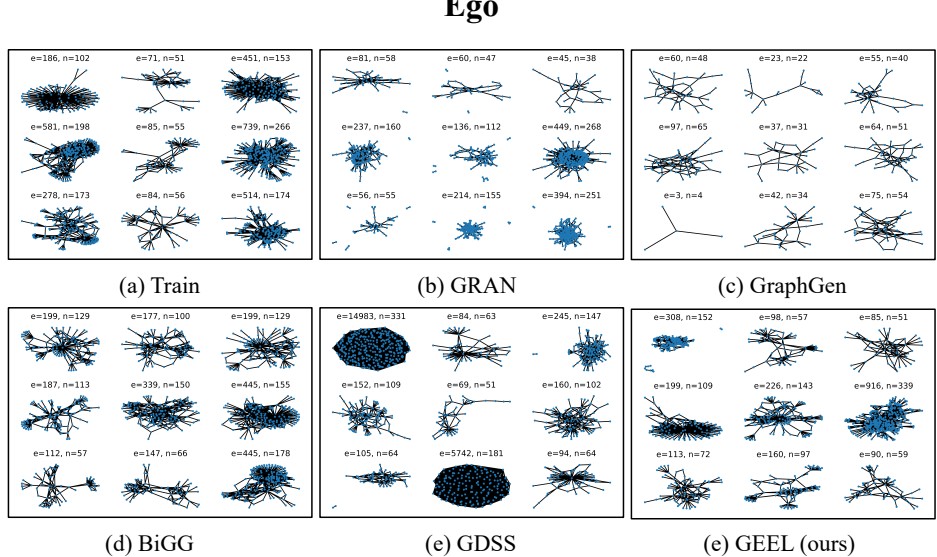

Figure 11: **Visualization of the graphs from the Ego dataset and the generated graphs.**

**Grid**

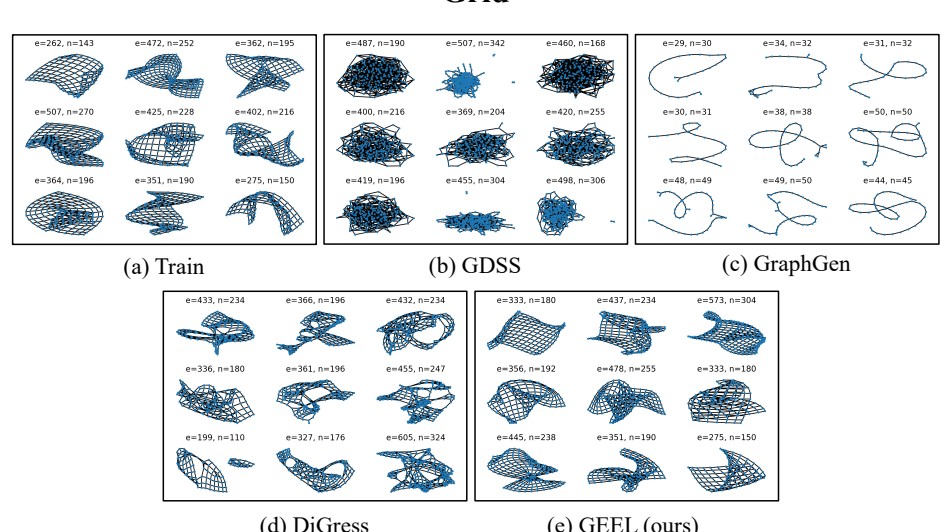

Figure 12: **Visualization of the graphs from the Grid dataset and the generated graphs.**

**Proteins**

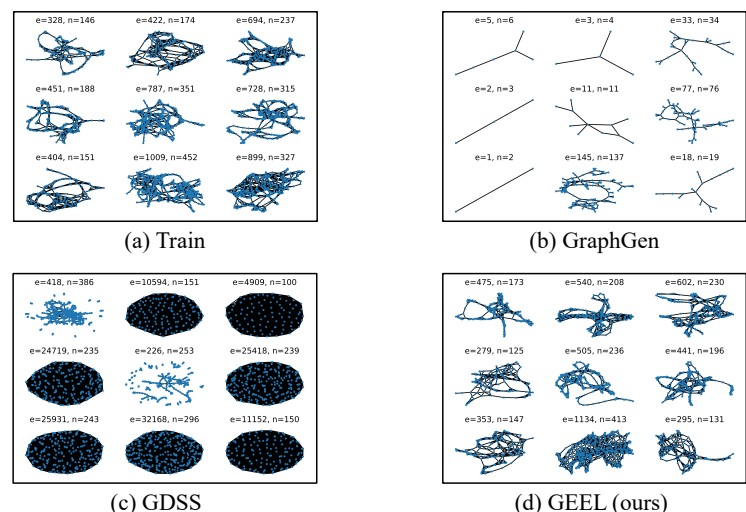

Figure 13: **Visualization of the graphs from the Proteins dataset and the generated graphs.**

We present visualizations of graphs from the training dataset and generated samples from GRAN, GraphGen, BiGG, GDSS, DiGress, and GEEL in Figure 7, Figure 8, Figure 9, Figure 10, Figure 11, Figure 12, and Figure 13. Note that we only provide the visualization that we have reproduced, which is detailed in Appendix B. We additionally give the number of nodes and edges of each graph, where $n$ denotes the number of nodes and $e$ denotes the number of edges.

### D.2 MOLECULAR GRAPH GENEREATION

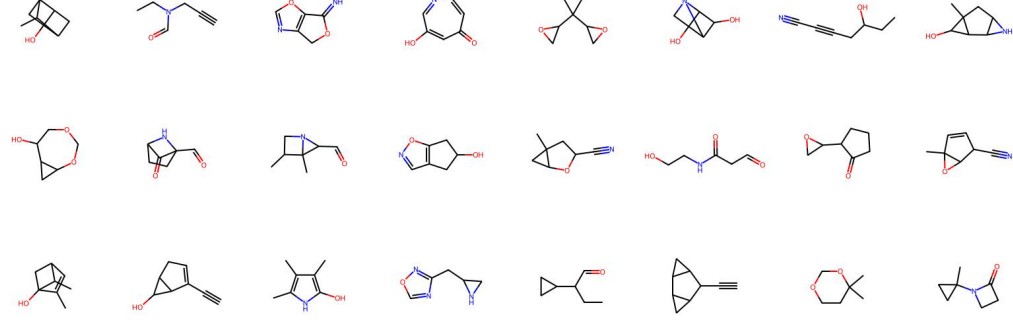

Figure 14: **Visualization of the molecules generated from the QM9 dataset.**

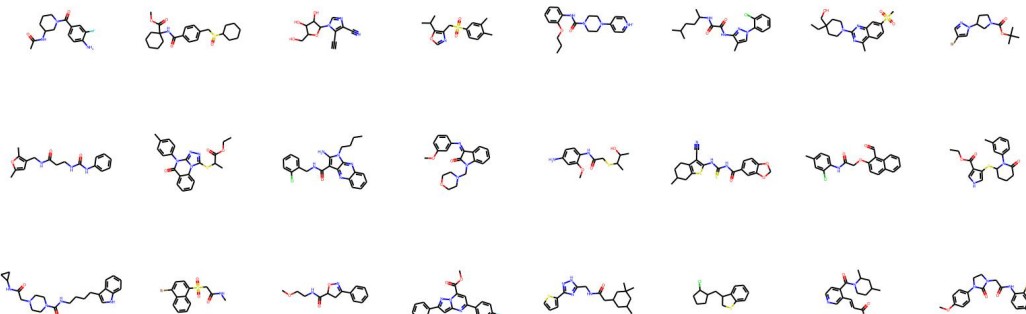

Figure 15: **Visualization of the molecules generated from the ZINC250k dataset.**

We provide visualizations of generated molecules using GEEL in Figure 14 and Figure 15.

# E  ADDITIONAL EXPERIMENTAL RESULTS

## E.1  GENERAL GRAPH GENERATION

Table 13: General graph generation on small graphs ($|V|_{\max} \leq 20$)

| | Ego-small | | | Community-small | | |
|---|---|---|---|---|---|---|
| | $4 \leq |V| \leq 18$ | | | $12 \leq |V| \leq 20$ | | |
| Method | Deg. | Clus. | Orb. | Deg. | Clus. | Orb. |
| Training | 0.025 | 0.035 | 0.012 | 0.020 | 0.044 | 0.003 |
| GraphVAE | 0.130 | 0.170 | 0.050 | 0.350 | 0.980 | 0.540 |
| GraphRNN | 0.090 | 0.220 | **0.003** | 0.080 | 0.120 | 0.040 |
| GRAN | **0.009** | **0.038** | **0.009** | **0.005** | 0.142 | 0.090 |
| GNF | **0.030** | 0.100 | **0.001** | 0.200 | 0.200 | 0.110 |
| EDP-GNN | 0.052 | 0.093 | **0.007** | 0.053 | 0.144 | 0.026 |
| GraphGen | 0.085 | 0.102 | 0.425 | 0.075 | 0.065 | 0.014 |
| GraphAF | **0.030** | 0.110 | **0.001** | 0.180 | 0.200 | 0.020 |
| GraphDF | 0.040 | 0.130 | **0.010** | 0.060 | 0.120 | 0.030 |
| BiGG | **0.013** | **0.030** | **0.005** | **0.004** | **0.005** | **0.000** |
| GDSS | **0.021** | **0.024** | **0.007** | 0.045 | 0.086 | 0.007 |
| DiGress | 0.021 | 0.026 | 0.024 | 0.012 | 0.025 | 0.002 |
| GDSM | - | - | - | **0.011** | **0.015** | **0.001** |
| GraphARM | **0.019** | **0.017** | **0.010** | 0.034 | 0.082 | **0.004** |
| SwinGNN | **0.000** | **0.021** | **0.004** | **0.003** | **0.051** | **0.004** |
| GEEL (ours) | **0.020** | **0.035** | **0.012** | **0.020** | **0.022** | 0.006 |

We provide general graph generation results for smaller graph datasets: Ego-small and Community-small. The **Ego-small** dataset consists of 300 small ego graphs from larger Citeseer network (Sen et al., 2008) and **Community-small** dataset consists of 100 randomly generated community graphs. We used the same split with GDSS (Jo et al., 2022) and the results are reported in Table 13.

## E.2  MOLECULAR GRAPH GENERATION

We provide molecular graph generation results for MOSES benchmark (Polykovskiy et al., 2020) in Table 14.

Table 14: **Molecular graph generation performance of MOSES dataset**. The baseline results are from prior works (Polykovskiy et al., 2020; Ahn et al., 2022). The best results are highlighted in **bold** and the second best are underlined.

| | MOSES | | | | | |
|---|---|---|---|---|---|---|
| Method | Validity (%) ↑ | NSPDK ↓ | FCD ↓ | Scaf. ↑ | SNN ↑ | Frag. ↑ |
| Molecule generative models | | | | | | |
| CharRNN | 97.48 | - | 0.0732 | 0.9242 | 0.6015 | **0.9998** |
| JT-VAE | **100.00** | - | 0.3954 | 0.8964 | 0.5477 | 0.9965 |
| STGG | **100.00** | - | **0.0680** | 0.9416 | 0.6359 | **0.9998** |
| Domain-agnostic graph generative models | | | | | | |
| DiGress | **100.00** | 0.0005 | 0.6788 | **0.8653** | 0.5488 | 0.9988 |
| GEEL (Ours) | 99.99 | **0.0001** | **0.1603** | 0.8622 | **0.6310** | **0.9996** |

# F    ADDITIONAL METRICS

## F.1    GENERAL GRAPH GENERATION

We provide three additional metrics: validity, uniqueness, and novelty scores for general graph generation in Table 15. GEEL achieves from 30% to 90% novelty, which is better than autoregressive models like GRAN and BiGG, but lower than the diffusion-based graph generative models. This is partially due to the inherent trade-off between novelty and the capability of generative models to learn the data distribution faithfully. OOM indicates out-of-memory and N.A. for BwR indicates that the generated samples are all invalid.

## F.2 MOLECULAR GRAPH GENERATION

We also provide the uniqueness and novelty scores for QM9 and ZINC in Table 16. We can observe that our GEEL shows competitive novelty and uniqueness compared to the baselines. Notably, the models make a tradeoff between the quality (e.g., NSPDK and FCD) and novelty of the generated graph since the graph generative models that faithfully learn the distribution put a high likelihood on the training dataset. In particular, the tradeoff is more significant in QM9 due to the large dataset size (134k) compared to the relatively small search space (molecular graphs with only up to nine heavy atoms).

Table 15: **The results of general graph generation include validity, uniqueness, and novelty.** The baseline results are from prior works (Martinkus et al., 2022; Vignac et al., 2022) or obtained by running the open-source codes.

| | Planar | | | | | | Lobster | | | | | |
| | $|V| = 64$ | | | | | | $10 \leq |V| \leq 100$ | | | | | |
| Method | Deg. (↓) | Clu. (↓) | Orb. (↓) | Val. (↑) | Uniq. (↑) | Nov. (↑) | Deg. (↓) | Clu. (↓) | Orb. (↓) | Val. (↑) | Uniq. (↑) | Nov. (↑) |
|---|---|---|---|---|---|---|---|---|---|---|---|---|
| GraphRNN | 0.005 | 0.278 | 1.254 | 0.0 | 100.0 | 100.0 | 0.000 | 0.000 | 0.000 | - | - | - |
| GRAN | 0.001 | 0.043 | 0.001 | 97.5 | 85.0 | 2.5 | 0.038 | 0.000 | 0.001 | - | - | - |
| SPECTRE | 0.001 | 0.079 | 0.001 | 25.0 | 100.0 | 100.0 | - | - | - | - | - | - |
| BiGG | 0.002 | 0.004 | 0.000 | 100.0 | 85.0 | 0.0 | 0.000 | 0.000 | 0.000 | - | - | - |
| GDSS | 0.250 | 0.393 | 0.587 | 0.0 | 100.0 | 100.0 | 0.117 | 0.002 | 0.149 | 18.2 | 100.0 | 100.0 |
| DiGress | 0.000 | 0.002 | 0.008 | 85.0 | 100.0 | 100.0 | 0.021 | 0.000 | 0.004 | 54.5 | 100.0 | 100.0 |
| BwR + GraphRNN | 0.609 | 0.542 | 0.097 | 52.5 | 95.0 | 100.0 | 0.316 | 0.000 | 0.247 | 100.0 | 63.6 | 100.0 |
| BwR + Graphite | 0.971 | 0.562 | 0.636 | 3.4 | 100.0 | 100.0 | 0.076 | 1.075 | 0.060 | 0.0 | 100.0 | 100.0 |
| BwR + EDP-GNN | 1.127 | 1.032 | 0.066 | 0.0 | 100.0 | 100.0 | 0.237 | 0.062 | 0.166 | 0.0 | 100.0 | 100.0 |
| GEEL (ours) | 0.001 | 0.010 | 0.001 | 82.5 | 97.5 | 27.5 | 0.002 | 0.000 | 0.001 | 72.7 | 100 | 72.7 |
| GEEL + No PE | 0.002 | 0.006 | 0.001 | 92.5 | 92.5 | 15.0 | 0.007 | 0.001 | 0.006 | 81.8 | 100.0 | 81.8 |

| | Enzymes | | | | | SBM | | | | |
| | $10 \leq |V| \leq 125$ | | | | | $31 \leq |V| \leq 187$ | | | | |
| Method | Deg. (↓) | Clu. (↓) | Orb. (↓) | Uniq. (↑) | Nov. (↑) | Deg. (↓) | Clu. (↓) | Orb. (↓) | Uniq. (↑) | Nov. (↑) |
|---|---|---|---|---|---|---|---|---|---|---|
| GraphRNN | 0.017 | 0.062 | 0.046 | - | - | 0.006 | 0.058 | 0.079 | 100.0 | 100.0 |
| GRAN | 0.023 | 0.031 | 0.169 | 100.0 | 94.9 | 0.011 | 0.055 | 0.054 | 100.0 | 100.0 |
| SPECTRE | - | - | - | - | - | 0.002 | 0.052 | 0.041 | 100.0 | 100.0 |
| BiGG | 0.010 | 0.018 | 0.011 | 78.8 | 4.2 | 0.029 | 0.003 | 0.036 | 92.5 | 10.0 |
| GDSS | 0.026 | 0.061 | 0.009 | - | - | 0.496 | 0.456 | 0.717 | 100.0 | 100.0 |
| DiGress | 0.011 | 0.039 | 0.010 | 100.0 | 99.2 | 0.006 | 0.051 | 0.058 | 100.0 | 100.0 |
| BwR + GraphRNN | 0.021 | 0.095 | 0.025 | 97.5 | 100.0 | N.A. | N.A. | N.A. | N.A. | N.A. |
| BwR + Graphite | 0.213 | 0.270 | 0.056 | 100.0 | 100.0 | 1.305 | 1.341 | 1.056 | 100.0 | 100.0 |
| BwR + EDP-GNN | 0.253 | 0.118 | 0.168 | 100.0 | 100.0 | 0.657 | 1.679 | 0.275 | 100.0 | 100.0 |
| GEEL (ours) | 0.005 | 0.018 | 0.006 | 100.0 | 94.9 | 0.025 | 0.003 | 0.026 | 95.0 | 42.5 |
| GEEL + No PE | 0.005 | 0.014 | 0.002 | 100.0 | 93.2 | 0.013 | 0.002 | 0.028 | 100.0 | 35.0 |

| | Ego | | | | | Proteins | | | | |
| | $50 \leq |V| \leq 399$ | | | | | $13 \leq |V| \leq 1575$ | | | | |
| Method | Deg. (↓) | Clu. (↓) | Orb. (↓) | Uniq. (↑) | Nov. (↑) | Deg. (↓) | Clu. (↓) | Orb. (↓) | Uniq. (↑) | Nov. (↑) |
|---|---|---|---|---|---|---|---|---|---|---|
| GraphRNN | 0.117 | 0.615 | 0.043 | - | - | 0.011 | 0.140 | 0.880 | 100.0 | 100.0 |
| GRAN | 0.026 | 0.342 | 0.254 | - | - | 0.002 | 0.049 | 0.130 | 100.0 | 100.0 |
| SPECTRE | - | - | - | - | - | 0.013 | 0.047 | 0.029 | 100.0 | 100.0 |
| BiGG | 0.010 | 0.017 | 0.012 | 89.5 | 57.2 | 0.001 | 0.026 | 0.023 | - | - |
| GDSS | 0.393 | 0.873 | 0.209 | 100.0 | 100.0 | 0.703 | 1.444 | 0.410 | 99.5 | 100.0 |
| DiGress | 0.063 | 0.031 | 0.024 | 100.0 | 100.0 | OOM | OOM | OOM | OOM | OOM |
| BwR + GraphRNN | N.A. | N.A. | N.A. | N.A. | N.A. | 0.092 | 0.229 | 0.489 | - | - |
| BwR + Graphite | 0.229 | 0.123 | 0.054 | 100.0 | 100.0 | 0.239 | 0.245 | 0.492 | - | - |
| BwR + EDP-GNN | OOM | OOM | OOM | OOM | OOM | 0.184 | 0.208 | 0.738 | - | - |
| GEEL (ours) | 0.053 | 0.017 | 0.016 | 89.4 | 62.8 | 0.003 | 0.005 | 0.003 | 93.7 | 88.9 |

| | 3d point cloud | | | | |
| | $8 \leq |V| \leq 5037$ | | | | |
| Method | Deg. (↓) | Clu. (↓) | Orb. (↓) | Uniq. (↑) | Nov. (↑) |
|---|---|---|---|---|---|
| GraphVAE | OOM | OOM | OOM | OOM | OOM |
| GraphRNN | OOM | OOM | OOM | OOM | OOM |
| GRAN | 0.018 | 0.510 | 0.210 | - | - |
| BiGG | 0.003 | 0.210 | 0.007 | - | - |
| GDSS | OOM | OOM | OOM | OOM | OOM |
| DiGress | OOM | OOM | OOM | OOM | OOM |
| BwR + GraphRNN | 1.820 | 1.295 | 0.869 | 100.0 | 100.0 |
| BwR + Graphite | OOM | OOM | OOM | OOM | OOM |
| BwR + EDP-GNN | OOM | OOM | OOM | OOM | OOM |
| GEEL (ours) | 0.002 | 0.081 | 0.020 | 100.0 | 80.0 |

Table 16: **Molecular graph generation performance of the QM9 and ZINC datasets including novelty and uniqueness.** The baseline results are from prior works (Jo et al., 2023; Ahn et al., 2022). The best results of molecule generative models and domain-agnostic generative models are both highlighted in **bold**.

| | QM9 | | | | | | | |
|---|---|---|---|---|---|---|---|---|
| Method | Validity (%) (↑) | NSPDK (↓) | FCD (↓) | Scaf. (↑) | SNN (↑) | Frag. (↑) | Unique. (%) (↑) | Novelty (%) (↑) |
| Molecule-specific generative models | | | | | | | | |
| CharRNN | 99.57 | **0.0003** | **0.087** | 0.9313 | 0.5162 | 0.9887 | - | - |
| CG-VAE | **100.0** | - | 1.852 | 0.6628 | 0.3940 | 0.9484 | - | - |
| MoFlow | 91.36 | 0.0169 | 4.467 | 0.1447 | 0.3152 | 0.6991 | **98.65** | **94.72** |
| STGG | **100.0** | - | 0.585 | 0.9416 | **0.9998** | **0.9984** | 96.76 | 72.73 |
| Domain-agnostic graph generative models | | | | | | | | |
| EDP-GNN | 47.52 | 0.0046 | 2.680 | 0.3270 | 0.5265 | 0.8313 | **99.25** | 86.58 |
| GraphAF | 74.43 | 0.0207 | 5.625 | 0.3046 | 0.4040 | 0.8319 | 88.64 | 86.59 |
| GraphDF | 93.88 | 0.0636 | 10.928 | 0.0978 | 0.2948 | 0.4370 | 98.58 | **98.54** |
| GDSS | 95.72 | 0.0033 | 2.900 | 0.6983 | 0.3951 | 0.9224 | 98.46 | 86.27 |
| DiGress | 98.19 | 0.0003 | 0.095 | 0.9353 | 0.5263 | 0.0023 | 96.67 | 25.58 |
| DruM | 99.69 | **0.0002** | 0.108 | **0.9449** | **0.5272** | 0.9867 | 96.90 | 24.15 |
| GraphARM | 90.25 | 0.0020 | 1.220 | - | - | - | - | - |
| GEEL (ours) | **100.0** | **0.0002** | **0.089** | 0.9386 | 0.5161 | **0.9891** | 96.08 | 22.30 |

| | ZINC | | | | | | | |
|---|---|---|---|---|---|---|---|---|
| Method | Validity (%) (↑) | NSPDK (↓) | FCD (↓) | Scaf. (↑) | SNN (↑) | Frag. (↑) | Unique. (%) (↑) | Novelty (%) (↑) |
| Molecule-specific generative models | | | | | | | | |
| CharRNN | 6.95 | **0.0003** | 0.474 | 0.4024 | 0.3965 | **0.9988** | - | - |
| CG-VAE | **100.0** | - | 11.335 | 0.2411 | 0.2656 | 0.8118 | - | - |
| MoFlow | 91.36 | 0.0169 | 4.467 | 0.1447 | 0.3152 | 0.6991 | **99.99** | **100.00** |
| STGG | 63.11 | 0.0455 | 20.931 | 0.0133 | 0.2352 | 0.7508 | **99.99** | 99.89 |
| Domain-agnostic graph generative models | | | | | | | | |
| EDP-GNN | 63.11 | 0.0485 | 16.737 | 0.0000 | 0.0815 | 0.0000 | 99.79 | **100.00** |
| GraphAF | 68.47 | 0.0442 | 16.023 | 0.0672 | 0.2422 | 0.5348 | 98.64 | 99.99 |
| GraphDF | 90.61 | 0.1770 | 33.546 | 0.0000 | 0.1722 | 0.2049 | 99.63 | **100.00** |
| GDSS | 97.01 | 0.0195 | 14.656 | 0.0467 | 0.2789 | 0.8138 | 99.64 | **100.00** |
| DiGress | 94.99 | 0.0021 | 3.482 | 0.4163 | 0.3457 | 0.9679 | **99.97** | 99.99 |
| DruM | 98.65 | **0.0015** | 2.257 | 0.5299 | 0.3650 | 0.9777 | **99.97** | 99.98 |
| GraphARM | 88.23 | 0.0550 | 16.260 | - | - | - | - | - |
| GEEL (ours) | **99.31** | 0.0068 | **0.401** | **0.5565** | **0.4473** | 0.992 | **99.97** | 99.89 |

# G DISCUSSION

## G.1 LIMITATION

In this section, we discuss the limitations of our GEEL. The limitation of GEEL is two-fold: (1) unable to generate graphs with unseen tokens and (2) dependency on the bandwidth. Since GEEL is based on the vocabulary with gap-encoded edge tokens, the model cannot generate the graphs with unseen tokens, i.e., the graphs with larger bandwidth than the training graphs. Nonetheless, we believe the generalization capability of our GEEL is strong enough in practice. In all the experiments, we verified that our vocabulary obtained from the training dataset entirely captures the vocabulary required for generating graphs in the test dataset.

Another limitation is the dependency on the bandwidth, i.e., the vocabulary size of GEEL is the square of the bandwidth. It is true that the vocabulary size of GEEL is highly dependent on the bandwidth. However, most of the real-world large-scale graphs have small bandwidths as described in Table 12 so $B$ being as large as $N$ is rare in practice. Additionally, for larger graphs with $B \approx N$, one could consider (a) decomposing the graph into subgraphs with small bandwidth and (b) separately generating the subgraphs using GEEL. This would be an interesting future avenue of research.

## G.2 COMPARISON TO BWR

Here, we compare our GEEL to BwR (Diamant et al., 2023). Our work proposes a new edge list representation with the vocabulary size of $B^2$ with gap encoding. Although we promote reducing the bandwidth $B$ using the C-M ordering, as proposed by BwR, our key idea, the gap encoding edge list is orthogonal to BwR. Our analysis of $B^2$ vocabulary size simply follows the definition of graph bandwidth. Moreover, one could also choose any bandwidth minimization algorithm (e.g., BFS ordering used in GraphRNN) to reduce the vocabulary size $B^2$.

In detail, we compare our GEEL to three variants of BwR. First, BwR+GraphRNN iteratively adds a node to the graph by adding the neighbors independently at once using a multivariate Bernoulli distribution. This ignores the dependencies between adjacent edges, which our GEEL captures via the edge-wise updates. Next, BwR+Graphite is a VAE-based generative model that generates the whole graph in a one-shot manner. This also does not consider the edge dependencies during generation, while our GEEL does. Finally, BwR+EDP-GNN is a diffusion-based model that generates the continuous relaxation of the adjacency matrix. It requires the generative model to learn the complicated reverse diffusion process, which results in underfitting of the model. This issue is especially significant since the BwR+EDP-GNN uses a small number of 200 diffusion steps.

## H    ADDITIONAL RESULTS FOR ABLATION STUDY

|        | Deg.  | Clus. | Orb.  |
|--------|-------|-------|-------|
| BFS    | 0.000 | 0.000 | 0.000 |
| DFS    | 0.000 | 0.000 | 0.000 |
| Random | 0.000 | 0.000 | 0.000 |
| C-M    | 0.000 | 0.000 | 0.000 |

Table 17: **MMD on different node orderings**

We provide the MMD performance of the ablation study on different orderings of Section 4.3 in Table 17. We can observe that the generation with any ordering eventually converges to the same levels of MMD.

