# OpenReview forum: "A Simple and Scalable Representation for Graph Generation"
_ICLR.cc/2024/Conference — ICLR 2024 poster_

### Official Review · Reviewer_aPij · 2023-10-30

**Soundness:** 2 fair
**Presentation:** 2 fair
**Contribution:** 2 fair
**Rating:** 5
**Confidence:** 4

**Summary:**

This paper proposes a molecule graph generation model GEEL, whose backbone can be LSTM or Transformer. GEEL is scalable to relatively large graphs in molecule generation. GEEL reduces the vocabulary size of the edge list representation by using intra- and inter-edge gap encodings. This proposed edge encoding method is novel.

**Strengths:**

1. It is good to have a table like Table 9 to show reproduced datasets.
2. The intra- and inter-edge gap encoding is delicate and useful.
3. The use of LSTM reduces the time complexity.

**Weaknesses:**

1. There is a problem in Appendix C.1 Table 11 (b) Large graphs (|V|max ≤ 187).

2. For the molecule generation task, it is important to generate some novel molecules for further filtering in drag design and other real-world scenarios. This paper does not evaluate the portion of the novel-generated molecules in Table 4. In the extreme case, the model may only be able to generate molecules in the training set.

3. It is hard to claim that GEEL outperforms BiGG. In Table 1, the performance is close. In terms of speed, BiGG runs on GeForce GTX 1080 Ti while GEEL runs on GeForce RTX 3090, which makes Table 2’s result unfair.

**Questions:**

If this paper includes some ablation study of replacing the intra- and inter-edge gap encoding with traditional encodings, it will be better. Table 6 could include more experiment results. The parameter size can be adjusted to avoid Out-Of-Memory.

---

> ### Author Response · Authors · 2023-11-17
>
> Dear reviewer aPij,
>
> We sincerely appreciate your comments and efforts in reviewing our paper. We think our paper has much improved in terms of clarity and rigorous evaluation, thanks to your comment. We especially found your comment **W3** to be critical to correct our unclear description of the computational environment!
>
> We address your question as follows. We also updated our manuscript which is highlighted in $\color{red}{\text{red}}$.
>
> ---
>
> **W1. There is a problem in Appendix C.1 Table 11 (b) Large graphs (|V|max ≤ 187).**
>
> Thank you for pointing out the typo. We fixed this to $399 \leq |V|_{\max} \leq 5037$ in our manuscript.
>
> ---
>
> **W2. Provide the novelty and uniqueness rates achieved by GEEL in the molecular generation experiments.**
>
> Thank you for pointing out additional metrics for molecular graph generation. We provide the novelty and uniqueness in the following table and updated the performance of molecular graph generation in our manuscript. One can observe that our GEEL shows competitive novelty and uniqueness compared to the baselines. We updated Table 16 in Appendix F of our manuscript.
>
>
> |                 |             |   QM9 |       |  ZINC |        |
> | --------------- | ----------- | -----:| -----:| -----:| ------:|
> |                 |             | Uniq. |  Nov. | Uniq. |   Nov. |
> | Molecule        | MoFlow      | 98.65 | 94.72 | 99.99 | 100.00 |
> |                 | STGG        | 96.76 | 72.73 | 99.99 |  99.89 |
> | Domain-agnostic | DiGress     | 96.67 | 25.58 | 99.97 |  99.89 |
> |                 | DruM        | 96.90 | 24.15 | 99.97 |  99.99 |
> |                 | GEEL (Ours) | 96.08 | 22.30 | 99.97 |  99.98 |
>
> We note that the models make a tradeoff between the quality (e.g., NSPDK and FCD) and novelty of the generated graph since the graph generative models that faithfully learn the distribution put a high likelihood on the training dataset. In particular, the tradeoff is more significant in QM9 due to the large dataset size (134k) compared to the relatively small search space (molecular graphs with only up to nine heavy atoms). We also note that one can compensate for the non-unique and non-novel molecules by filtering them out and generating new molecules.
>
> ---
> **W3. It is hard to claim that GEEL outperforms BiGG. In Table 1, the performance is close. In terms of speed, BiGG runs on GeForce GTX 1080 Ti while GEEL runs on GeForce RTX 3090, which makes Table 2’s result unfair.**
>
> Thank you for pointing out the unclarity. We conducted the inference time analysis in Table 2 on the same computational environment (GTX 1080 Ti) for all the models including GEEL and BiGG. We have clarified this in our updated manuscript in Appendix B.3.
>
> Hence, one can safely conclude that GEEL is faster than BiGG. Furthermore, we claim that GEEL slightly outperforms BiGG in Table 1 since GEEL is marked bold for 20 out of 24 metrics in Table 1, while BiGG is marked bold for 14 out of 24 metrics. As mentioned in Section 4.1, we mark the numbers in bold if it is comparable to the metrics of the training graphs.
>
> ---
>
> **Q1. If this paper includes some ablation study of replacing the intra- and inter-edge gap encoding with traditional encodings, it will be better. Table 6 could include more experiment results. The parameter size can be adjusted to avoid Out-Of-Memory.**
>
> The ablation study of replacing the intra- and inter-edge gap with traditional encodings (i.e., original edge list) is in Table 6.  The edge list (second row) indicates the traditional encodings and the edge list + intra gap (third row) indicates employing only intra-edge gap. We updated our manuscript to clarify what each row in Table 6 means.
>
> Notably, the traditional encoding encounters out-of-memory in our computational environment even after adjusting the parameters or even setting the batch size to 1. This supports the superiority of our GEEL, concerning the memory requirement.

---

> ### Comment · Reviewer_aPij · 2023-11-22
>
> I have read the responses from the authors. I maintain my rating. The responses only addressed some of my concerns.
>
> BTW, it should be "Related Work", instead of "Related Works".

---

> ### Author Response · Authors · 2023-11-22
>
> Thank you for acknowledging our response and the efforts to review our paper.
>
> We would be grateful if you could let us know your unresolved concern. The concern might be something we could resolve easily, e.g., miscommunication during the rebuttal phase. It would also help us improve our work.
>
> ~~We were unsure of what you meant by *it should be "Related Work", instead of "Related Work".*,  but we changed our "Related Works" to "Related Work" section. Please let us know if we wrongly interpreted your comment.~~

---

### Official Review · Reviewer_RDfu · 2023-10-31

**Soundness:** 2 fair
**Presentation:** 3 good
**Contribution:** 3 good
**Rating:** 6
**Confidence:** 3

**Summary:**

Good paper, extensive evaluations.

The paper proposes a simple and scalable graph representation i.e  gap encoded edge list (GEEL) for graph generative modeling.
The  representation size  aligns with the number of edges instead of nodes, consequently low size and more applicable for sparse graphs.

There do exists works which use O(#edges) for representation, however, the proposed work GEEL further reduces the edge list representation by reducing vocabulary size from N^2 to B^2, where B is graph bandwidth.

The authors extend their approach to attributed graphs(having labels).

Empirical evaluation is performed on a large number of attributed and non-attributed datasets on a diverse set of graph similarly metrics.
The approach also scales better in terms of inference speed when compared to existing works.

Overall the approach shows significant improvements over existing methods for graph generative modeling tasks.

**Strengths:**

1. Novel approach for graph generative modeling-> compact representation of data.

2. Paper is easy to read. The authors clarify the need of each component. Diagrams are provided for better understanding of the method.

3. Ablation studies are conducted to understand  impact of different sequence encoding(DFS, BFS etc.), diff architectures for sequence modeling such as LSTM, Transformers etc.

4.High reproducibility: The experimental section is very detailed  with respect to the current work as well as baselines. Code is also provided.

5. The authors also visualize generated graphs.

Significance:
The proposed method could pave way for advancing research in context of graph generative modeling especially w.r.t to larger graphs( which are also attributed).

**Weaknesses:**

I would point to the questions section for details

1. Not clear how many graphs are generated for comparison.

2. Results on uniqueness and novelty metric seem to be missing.

3. Scalability analysis doesn't seem to be complete.

I request the authors to look into the questions section for details on each of the above point.

**Questions:**

1. It is not clear or I could not find out how many graphs were generated by the proposed method/ baselines for each dataset? I agree MMD is computed but how many graphs were generated? Request the authors to add this.

2. Results on Uniqueness and Novelty seem to be missing. It is not clear whether the generated graphs have duplicacy etc.
Refer metrics section of [A] for details.
Adding these metrics(atleast for few datasets) could further improve the quality of the manuscript.

3. Could the authors clarify how scalaibiltiy results are computed? I mean is batch size etc. set to 1?
Also since GraphGen[A] also works at edge list level O(#Edges), I would expect the authors to compare with GraphGen for scalability comparison.
Can the authors justify why generation time is shown for one graph? Could it be an outlier?  I see only one number without any standard deviation etc.
Would it make sense to generate a batch of graphs and report mean+std dev.





[A]Goyal et al. GraphGen: A Scalable Approach to Domain-agnostic Labeled Graph Generation, WWW 2020

---

> ### Author Response · Authors · 2023-11-17
>
> Dear reviewer RDfu,
>
> We sincerely appreciate your comments and efforts in reviewing our paper. We address your question as follows. We also updated our manuscript which is highlighted in $\color{red}{\text{red}}$. We think our paper has much improved in terms of clarity and rigorous evaluation, thanks to your comment.
>
> ---
>
> **W1/Q1. It is not clear how many graphs were generated for evaluation.**
>
> Thank you for pointing out the unclarity. Following prior works [1, 2], we generated graphs with the same number of test graphs. We updated our manuscript to clarify this in Section 4.1.
>
> ---
>
> **W2/Q2. Results on uniqueness and novelty metrics seem to be missing.**
>
> Thank you for such an insightful comment! Following your suggestion, we evaluated the validity, novelty, and uniqueness of GEEL and the considered baselines. We report the new results in Table 15 and Table 16 of Appendix F. We think our empirical results are now much more comprehensive and reliable, thanks to your review.
>
> In our new experiments, GEEL achieves the 30\% \~ 90\% novelty, which is better than autoregressive models like GRAN and BiGG, but lower than the diffusion-based graph generative models as the reviewer anticipated. This is partially due to the inherent trade-off between novelty and the capability of generative models to faithfully learn the data distribution. Even diffusion models suffer from this trade-off, e.g., DiGress achieves 30\% novelty for the QM9 dataset. We also note that this trade-off is less severe for harder problems, e.g., point cloud and ZINC250k, which is the main target of our work.
>
> Finally, we note the minor modification of GEEL in the new experiments. Our GEEL now samples a new C-M ordering for a graph at each training step (instead of fixing a unique ordering per graph). We found this to improve novelty without a decrease in performance and changing the architecture. We updated all of our experimental results to incorporate this change. Note that the results for the Ego dataset are to be updated, as the experiments have not yet been completed.
>
>
>
> ---
>
> **W3/Q3-1. Could the authors clarify how scalability results are computed?**
>
> We conducted inference time analysis by setting the batch size to 10 and dividing the total inference time by 10 to compute the generation time for a single graph. Therefore, the reported inference time is not an outlier that shows the time for a single graph generation. We updated our manuscript to clarify this in Appendix B.
>
> **Q3-2. Since GraphGen works at the edge list level, I would expect the authors to compare with GraphGen in Table 6.**
>
> To incorporate your suggestion, we will add a comparison with GraphGen in our future manuscript. Please understand that we are unable to provide the result during the rebuttal phase. Unfortunately, we currently do not have the computational environment (GeForce GTX 1080 Ti) used for Table 6, which is necessary for a fair comparison.
>
>
> **References**
> [1] Jo, J., et al., Score-based generative modeling of graphs via the system of stochastic differential equations. ICML 2022.
>
> [2] Martinkus, K., et al., SPECTRE: Spectral conditioning helps to overcome the expressivity limits of one-shot graph generators. ICML 2022.

---

> ### Comment · Reviewer_RDfu · 2023-11-21
>
> I thank the authors for the detailed feedback and rebuttal.
>
> 1. I am happy the authors added additional results on novelty and uniqueness. The metrics don't seem to be very good. So does that mean, they are replicating training data and not generating diverse samples?
>
> 2. I appreciate the authors for clarifying the novelty w.r.t BWR baseline of ICML 2023.
>
> 3. I don't see any comparison with BWR baseline for any tables in the appendix and Tab 2 and 4 in main paper. Is there any specific reason?
>
>
> Can the authors clearly justify the gains their method has against BWR( in terms of quality and scalability demonstrating on multiple datasets under  ) to make the manuscript more clear.
>
>  I am not sure if the manuscript is ready for publication yet. If the authors can better clarify things, it would be better.
>
> I am changing my score to weak accept for now.

---

> > ### Author Response · Authors · 2023-11-22
> >
> > Thank you for responding to our rebuttal and expressing your concerns. We hope to alleviate your concerns in what follows. We also updated our manuscript to further clarify your concern and they are marked in $\color{teal}{\text{green}}$.
> >
> > **The novelty and the uniqueness metrics do not look very good.**
> >
> > Our novelty and uniqueness metrics may appear even worse due to the comparisons with the models with low MMD scores tend to exhibit high novelty and uniqueness, e.g., GDSS and GraphRNN. When compared with the main competitor for large-scale graph generation, i.e., BiGG, our GEEL shows better uniqueness and novelty scores.
> >
> > While the novelty metric does count the number of replications from training data, we would like to point out that this also implies how our model successfully learned the underlying pattern in the dataset, especially when the pattern is simple. For example in the QM9 dataset, our argument aligns with that made by the Digress [1] paper: *"QM9 is an exhaustive enumeration of the small molecules that satisfy a given set of constraints, generating molecules outside this set is not necessarily a good sign that the network has correctly captured the data distribution."*  We also point out that our uniqueness metrics are at least 93.7%, which is a reasonable score for practical usage of our algorithm.
> >
> > **Why not compare with BwR in the Appendix?**
> >
> > We did not compare with BwR in Table 15 in the Appendix since the table requires validity, uniqueness, and novelty scores, which BwR did not report. Nonetheless, to alleviate your concern, we added BwR to Table 15 using the official code for the Planar and Enzymes datasets. Our GEEL showed superior performance compared to all variants of BwR in MMD. Note that our reproduced results differ from the reported ones in the BwR paper as the numbers of graphs in the dataset are different (e.g., BwR used 1500 planar graphs while we used 200 planar graphs).
> >
> > In addition, the official BwR implementation uses a different number of graph samples for evaluating the metrics for BwR+GraphRNN. The common protocol (used by GRAN, GDSS, and Digress) is to generate the same number of graphs as the test dataset. However, BwR discards the invalid graphs and evaluates a smaller number of graphs. In contrast, our method and other baselines ensure the evaluation of the same number of graphs as the test dataset, even after filtering out the invalid graphs. Note that the difference is especially significant since BwR filters out a large portion of invalid graphs, e.g., only 20% of samples are left for grid graphs.
> >
> > Below is the code block of the BwR official repository that filters out invalid graphs (`graph_gen/models/graph_rnn.py`).
> >
> > ```
> > def evaluate_temperature(
> >         self, temp: float, real_graphs: list[nx.Graph],
> >         max_graph_len: int,
> >     ) -> dict[str, float]:
> >         samples = self.sample(len(real_graphs), steps=max_graph_len, temperature=temp)
> >         bf = BandFlatten(self.bw)
> >         sampled_graphs = []
> >         for sample in samples:
> >             sampled_no_start_token = sample[1:]
> >             stop_idx = sampled_no_start_token[:, 0].argmax()
> >             sampled_clean = sampled_no_start_token[:stop_idx, 1:].cpu()
> >             try:
> >                 graph = graph_from_bandflat(sampled_clean, bf)
> >             except ValueError:
> >                 continue
> >             sampled_graphs.append(graph)
> > ```
> >
> > Note that we also updated Table 1 for Planar, Enzymes, and Grid datasets. We are running additional experiments for the remaining datasets: lobster, sbm, ego, and 3d point cloud. We will keep updating the manuscript after the experiments end.

---

> > > ### Author Response · Authors · 2023-11-22
> > >
> > > **Why not compare with BwR in Table 2?**
> > >
> > > For Table 2, we originally did not compare with BwR due to its underwhelming MMD performance (similarly for GraphGen). We could not reproduce all the baselines under the same computational environment, i.e., BiGG required GTX 1080, and have chosen to prioritize ones with more meaningful performance.
> > >
> > > Nonetheless, to alleviate your concern, we provide additional analysis on the inference time below. Note that we use GTX 3090 Ti GPU for this setting, which is a different setting from Table 2 (GTX 1080).
> > >
> > > |              | Enzymes | Planar | Grid |
> > > | ------------ | ------- | ------ |:----:|
> > > | BwR+GraphRNN | 0.02    | 0.11   | 0.35 |
> > > | BwR+EDP-GNN  | 0.10    | 0.20   | 0.31 |
> > > | BwR+Graphite | 0.20    | 0.23   | 0.49 |
> > > | GEEL (Ours)  | 0.09    | 0.18   | 1.24 |
> > >
> > >
> > > One can observe that our GEEL is slower than the BwR variants for some cases, but we again point out that a direct comparison is not meaningful due to the underwhelming performance of BwR baselines.
> > >
> > > We further elaborate that our GEEL is slower than BwR variants since the BwR variants use a fewer number of neural network updates. To be specific, BwR+GraphRNN, BwR+EDP-GNN, and BwR+Graphite generate one graph in $N, 200, 1$ neural network updates, respectively. Here, $N$ is the number of vertices, $200$ is the number of reverse diffusion steps, and Graphite is a VAE-based approach that generates a graph in a one-shot manner.
> > >
> > > **Why not compare with BwR in Table 4?**
> > >
> > > For Table 4 (molecular graphs), we were unable to compare with BwR which only provides implementation for non-attributed graphs. The official repository of BwR experiments on molecules by first converting it into non-attributed graphs using the following code (`bandwidth-graph-generation/graph_gen/data/mol_utils.py`):
> > >
> > >
> > > ```
> > > def smiles_to_nx_no_features(sm: str) -> nx.Graph:
> > >     mol = Chem.MolFromSmiles(sm)
> > >     adj = Chem.GetAdjacencyMatrix(mol)
> > >     return nx.from_numpy_matrix(adj)
> > >
> > >
> > > def get_zinc_graphs(
> > >     zinc_path: str = os.path.join(DATASET_PATH, "zinc.tab"),
> > > ) -> list[nx.Graph]:
> > >     zinc = pd.read_csv(zinc_path)
> > >     zinc_graphs = list(map(smiles_to_nx_no_features, zinc["smiles"].values))
> > >     return zinc_graphs
> > > ```
> > >
> > >
> > > We also acknowledge that BwR reported experiments on the molecular datasets (QM9, ZINC250k). However, given the official implementation, it is unclear if the experiments were conducted on the attributed molecular graphs. The types of metrics, e.g., log-likelihood and AUPRC, are also quite different from our setting, hence we cannot directly compare them with the reported numbers.
> > >
> > > **Can the authors clearly justify the gains their method has against BWR (in terms of quality and scalability demonstrating on multiple datasets under) to make the manuscript more clear?**
> > >
> > > Thank you for the suggestion. Since our GEEL and the BwR variants share the search space, i.e., graphs with limited bandwidth, we believe the gains stem from our edge-by-edge generation with a small vocabulary size, which is our main contribution.
> > >
> > > In what follows, we compare with the BwR variants one by one.
> > > - BwR+GraphRNN iteratively adds a node to the graph by adding the neighbors *independently* at once using a multivariate Bernoulli distribution. This ignores the dependencies between edges being added at each step, which our GEEL captures via the edge-wise updates.
> > > - BwR+Graphite is a VAE that generates the whole graph in one-shot. This again does not consider the edge dependencies during generation, while our GEEL does.
> > > - BwR+EDP-GNN is a diffusion-based model that generates the continuous relaxation of the adjacency matrix. It requires the generative model to learn the complicated reverse diffusion process, which results in the underfitting of the model. This problem is especially significant since the BwR+EDP-GNN uses a small number of 200 diffusion steps.
> > >
> > > We have also incorporated the comparison in Appendix G.2. our updated manuscript. We believe our gains are more clearly justified, thanks to your comments.
> > >
> > > **References**
> > >
> > > [1] Vignac, C., et al., DiGress: Discrete Denoising diffusion for graph generation, ICLR 2023.

---

> ### Comment · Reviewer_RDfu · 2023-11-22
> **Thank you :)**
>
> I thank the authors for the clarifications.
> Most of my concerns have been resolved.
>
> I keep my score to 6 and would recommend acceptance of the paper for their interesting ideas
> Overall it's a good paper :).

---

> > ### Author Response · Authors · 2023-11-22
> >
> > Thank you for acknowledging our response and the efforts in reviewing our paper. We believe your reviews were very helpful in improving our paper during the rebuttal process. Many thanks!

---

### Official Review · Reviewer_kVSU · 2023-11-01

**Soundness:** 3 good
**Presentation:** 4 excellent
**Contribution:** 3 good
**Rating:** 8
**Confidence:** 4

**Summary:**

This paper proposes GEEL, a novel method to generate graphs starting from their sorted edge list. In particular, the edge list is encoded through gaps: each pair of nodes representing an edge becomes a pair where the first element is the gap from the previous pair's first element, while the second element encodes the gap from the current pair's first element. This has the advantage of reducing the vocabulary size from $N^2$ (where $N$ is the number of graph nodes) to $B^2$ (where $B$ is the graph bandwidth), while maintaining the cost of training and inference to be $O(M)$ (where $M$ is the number of edges). The method is extended to deal with attributed graphs by proposing a simple grammar whose elements are triplets of the form (node type token, gap-encoded edge, edge type token), plus rules to compose them meaningfully.

The generative model is an auto-regressive LSTM that is trained to maximize the likelihood of the training graphs (represented as sequences of gap-encoded edge pairs). The model is evaluated extensively in task of generating a) standard non-attributed graphs such as lobster, ego, community and b) molecules. The experiments show that a) the proposed approach achieves good generative performance with respect to a wide pool of competitors, b) it uses a parsimonious representation which allows reduced vocabulary size and competitive training/inference cost. Lastly, ablation studies are presented to justify the architectural choices.

**Strengths:**

I am very pleased with this paper: it is written clearly and easy to follow. On the technical side, it presents a simple but empirically effective contribution, which addresses most of the challenges of edge-list based graph generative methods, namely the large vocabulary sizes and the burden of having to learn long-term dependencies as a consequence. Following, a list of things I identify as strengths:

- The main contribution (the gap-encoded edge list) is novel.
- The proposed approach is extremely simple yet very effective.
- The literature review is satisfactory (although it is missing [1] as another edge-list based generative method, but that is a minor omission).
- The experiments are thorough, spanning across different graph types and a not common wide range of baselines.
- I appreciated the ablation study which help understanding why certain modelling choices were taken.



[1] Bacciu et al., Edge-based sequential graph generation with recurrent neural networks. Neurocomputing 2020.

**Weaknesses:**

The weaknesses I have found are by no means fatal, and I believe they could be addressed through proper rebuttal. In particular:

- perhaps this article is not well-suited for ICLR, since the focus should be on learning representations, but this paper does not revolve around representation learning. Again, I don't think this is fatal, and I would like to hear how this work places itself in the context of this conference by the authors.

- while very effective, this method has also limitations which are not mentioned by the authors. The main one being the fact that it is still a vocabulary-based approach that cannot generalize to graphs with gap-encoded edges not present in the vocabulary. Another one is the dependency on the bandwidth $B$, which can sometimes be $\approx N$ due to outliers. I understand that this can be bypassed by removing the outliers, but then again it restricts the applicability of the method to a certain class of graphs (those with low bandwidth) to exploit the concise gap-encoding.

**Questions:**

I have some:

- What do the asterisks placed after Graphgen and Graphgen-redux in Table 1 mean?
- What is meant by "comparable" MMD? Which criteria is used to define two MMDs comparable?
- In Figure 4, what is the value of the $c_1$ and $c_2$ constants? Have they been explicitly calculated?
- What are the novelty and uniqueness rates achieved by GEEL in the molecular generation experiments? How do they compare to the competitors? For example, I recall CharRNN has a novelty rate of about 84% on the MOSES benchmark, while STGG has a novelty rate of 67% on the same benchmark.
- Speaking of which, I think it would be better to add the performances of the method in the MOSES benchmark. It should be fairly doable in a short time since the method is fast both in training and inference.
- What are the vocabulary sizes for the molecular generation experiments (on QM9 and ZINC250k)?
- Which order is used to encode the edge list of molecules? Is it the SMILES canonical order?
- If the answer to the question above is yes: as you might know, SMILES works by generating a spanning tree, and then adding the edges that close the rings at the end. Don't you think this kind of process is not ideal to gap-encode the edge list of molecules (since the second element of the closing ring edges would have an abnormally wider gap with respect to the other edges)?

**Details Of Ethics Concerns:**

None.

---

> ### Author Response · Authors · 2023-11-17
>
> Dear reviewer kVSU,
>
> We sincerely appreciate your comments and efforts in reviewing our paper. We think our paper has much improved in terms of clarity and positioning our work, thanks to your comment.
>
> We address your question as follows. We also updated our manuscript which is highlighted in $\color{red}{\text{red}}$.
>
> ---
>
> **W1. The work is perhaps not well-suited for ICLR since it does not include any learning representations.**
>
> Thank you for checking this. We would like to point out that ICLR 2024 mentions "generative models" as one of the relevant topics in its call for papers (https://iclr.cc/Conferences/2024/CallForPapers). Hence we believe our work (graph generative model) aligns with the context of ICLR.
>
> We also provide a few examples of similar works in previous ICLRs:
> - Shi, C., et al., GraphAF: a Flow-based Autoregressive Model for Molecular Graph Generation, ICLR 2020.
> - Liu, M., et al., GraphEBM: Molecular Graph Generation with Energy-Based Models, ICLR 2021.
> - Vignac, C., et al., DiGress: Discrete Denoising diffusion for graph generation, ICLR 2023.
>
> ---
>
> **W2-1. It is not mentioned that GEEL cannot generate graphs with unseen tokens, since it is a vocabulary-based approach.**
>
> We agree with your comment and mentioned this in our updated manuscript in Appendix G. Nonetheless, we believe the generalization capability of our GEEL is strong enough in practice. In all the experiments, we verified that our vocabulary obtained from the training dataset entirely captures the vocabulary required for generating graphs in the test dataset.
>
> **W2-2. It is not mentioned that GEEL depends on the bandwidth $B$, which can be $≈N$ for outliers.**
>
> We agree with your comment and mentioned this in our updated manuscript in Appendix G. As you mentioned, one could bypass this issue by removing the outlier graph or even removing the particular edge that increases the bandwidth. One also could consider (a) decomposing the graph into subgraphs with small bandwidth and (b) separately generating the subgraphs using GEEL. This would be an interesting future avenue of research.
>
> Finally, as the reviewer mentioned, we would like to re-emphasize how the case $N\approx B$ is quite rare in practice.
>
> ---
>
> **Q1. What do the asterisks placed after Graphgen and Graphgen-redux in Table 1 mean?**
>
> Thank you for pointing out this typo. The asterisks indicated edge list-based graph generative models, but we deleted them since they are not very informative. We instead mention that Graphgen and Graphgen-redux are edge list-based graph generation methods in Section 4.
>
> ---
>
> **Q2. What is meant by "comparable" MMD? Which criteria is used to define two MMDs comparable?**
>
> The comparability of MMD values is determined by examining whether the MMD falls within a range of one standard deviation (reported in Table 11 of Appendix C.1). This was explained in the evaluation protocol in Section 4.1, but we also added this to the caption of Table 1 to further be more clear.
>
> ---
>
> **Q3. In Figure 4, what is the value of the c1 and c2 constants? Have they been explicitly calculated?**
>
> The constants $c_1$ and $c_2$ in Figure 4 are 0.5 and 2, respectively. We added the lines for visual alignment between the inference time and the theoretical computational complexity $O(M)$, where $M$ is the number of edges. We tried a few values to select the visually plausible constants.
>
> ---
>
> **Q4. What are the novelty and uniqueness rates achieved by GEEL in the molecular generation experiments?**
>
> Thank you for pointing out additional metrics for molecular graph generation. We provide the novelty and uniqueness in the following table and updated the performance of molecular graph generation in our manuscript. We can observe that our GEEL shows competitive novelty and uniqueness compared to the baselines. We reported the new results in Table 16 of Appendix F.
>
> |                 |             |   QM9 |       |  ZINC |        |
> | --------------- | ----------- | -----:| -----:| -----:| ------:|
> |                 |             | Uniq. |  Nov. | Uniq. |   Nov. |
> **Molecule-specific**       | MoFlow      | 98.65 | 94.72 | 99.99 | 100.00 |
> |                 | STGG        | 96.76 | 72.73 | 99.99 |  99.89 |
> | **Domain-agnostic** | DiGress     | 96.67 | 25.58 | 99.97 |  99.98 |
> |                 | DruM        | 96.90 | 24.15 | 99.97 |  99.99 |
> |                 | GEEL (Ours) | 96.08 | 22.30 | 99.97 |  99.89 |
>
> We note that the models make a tradeoff between the quality (e.g., NSPDK and FCD) and novelty of the generated graph since the graph generative models that faithfully learn the distribution put a high likelihood on the training dataset. In particular, the tradeoff is more significant in QM9 due to the large dataset size (134k) compared to the relatively small search space (molecular graphs with only up to nine heavy atoms). We also note that one can compensate for the non-unique and non-novel molecules by filtering them out and generating new molecules.

---

> > ### Author Response · Authors · 2023-11-17
> >
> > **Q5. Add additional MOSES dataset for molecular graph generation.**
> >
> > Thank you for pointing out the meaningful benchmark. We provide the molecular generation performance on the MOSES dataset in the following table and will incorporate this into our future manuscript. We can observe that GEEL shows competitive performance to molecule-specific methods and superior performance to the domain-agnostic method, DiGress. We reported the new results in Table 14 of Appendix E.
> >
> >
> > |              |   Method    |     Val. ↑ | NSPDK ↓ |      FCD ↓ |    Scaf. ↑ |     SNN  ↑ |    Frag. ↑ | Uniq.  ↑ | Nov. ↑ |
> > |:------------:|:-----------:| ----------:| -------:| ----------:| ----------:| ----------:| ----------:| -------:| -------:|
> > |   **Molecule-specific**   |   CharRNN   |      97.48 |       - |     0.0732 |     0.9242 |     0.6015 |     0.9998 |    0.9909     |     0.8419    |
> > |              |   JT-VAE    | 100.00 |       - |     0.3954 |     0.8964 |     0.5477 |     0.9965 |    0.9996     |     0.9143    |
> > |              |    STGG     | 100.00 |       - | 0.0680 | 0.9416 | 0.6359 | 0.9998 |   0.9987      |     0.6727    |
> > | **Domain-agnostic** |   DiGress   | 100.00 |  0.0005 |     0.6788 |     0.8653 |     0.5488 |     0.9988 |    0.9999     |    0.9447     |
> > |              | GEEL (Ours) |    100.00        |    0.0001     |      0.1528      |      0.8692      |       0.6273     |     0.9996       |    0.9989     |  0.7313       |
> >
> >
> > ---
> >
> > **Q6. What are the vocabulary sizes for the molecular generation experiments (on QM9 and ZINC250k)?**
> >
> > The vocabulary size of the molecules is $2B+|X|+|E|$, where $B$ denotes the bandwidth, $|X|$ denotes the number of node features, and $|E|$ denotes the number of edge features. The bandwidths, the number of nodes, and the number of edges of QM9 and ZINC250k are provided in the following table, which are also provided in Appendix A and C in our original manuscript. The vocabulary sizes of QM9 and ZINC250k are 27 and 53, respectively. We updated our manuscript in Section 3.3.
> >
> >
> > |          | $B$ | $X$ | $E$ | Vocab. |
> > |:--------:|:---:|:---:|:---:|:------:|
> > |   **QM9**    |  5  | 13  |  4  |   27   |
> > | **ZINC250k** | 10  | 29  |  4  |   53   |
> >
> >
> > ---
> >
> > **Q7. Which order is used to encode the edge list of molecules? Is it the SMILES canonical order?**
> >
> > We used C-M ordering just the same as the general graphs. It is different from the SMILES canonical order as we generate graphs with gap-encoded edge list tokens, not with SMILES characters.

---

> > > ### Comment · Reviewer_kVSU · 2023-11-22
> > > **Thank you**
> > >
> > > I have read your responses, the overall conversations and the updated manuscript. I was aware of the existence of BwR, and I understand that the attribution of BwR was a bit unclear to anyone who didn't know about the method. I'm glad you fixed it and that is now clearly expressed. My concerns are resolved, so I'm raising my score to an 8. Good luck!

---

> > > > ### Author Response · Authors · 2023-11-22
> > > >
> > > > We greatly appreciate your positive response. Your comments were very helpful in enhancing our paper, and we are glad to hear that our efforts have addressed most of the concerns including the unclarity of BwR. Many thanks!

---

### Official Review · Reviewer_EmZF · 2023-11-01

**Soundness:** 3 good
**Presentation:** 3 good
**Contribution:** 3 good
**Rating:** 6
**Confidence:** 4

**Summary:**

Authors introduce a new parametrization for sequential generation of graphs. The parametrization is based on edge-lists with the main trick being encoding not the node IDs but difference between them. The use of C-M ordering is promoted and shown to be better than the usual DFS or BFS orderings used in the previous autoregressive models. Overall the proposed architecture is shown to be much faster than existing approaches and offer better or competitive results in graph generation quality.

**Strengths:**

The proposed representation is very interesting and efficient. It also meshes well with the C-M ordering.
The choices are extensively ablated and shown to be better than alternatives. The experimental performance overall seems strong and experimental scalability is good.
The paper is well written and easy to follow.

**Weaknesses:**

My main concern with the paper are the evaluation metrics reported for the graph generation. First, only local MMD scores are reported (e.g. no Spectral MMD as used in GRAN or SPECTRE), but mainly that the novelty and uniqueness of the generated examples is not reported. As shown in the SPECTRE paper, autoregressive models such as GRAN can overfit the training set to a point, where effectively no novel samples are produced, while at the same time producing amazing MMD metrics. Since this work is essentially turbocharging the autoregressive generation, especially with the relative node ID representation, one could imagine that such overfitting would be a problem. After all this overfitting is one of the main motivations for using one-shot generative models without a given node ordering. Uniqueness and novelty is also commonly reported for the molecule generation as well.

It would also be interesting to see how some of the one-shot methods (e.g. DiGress) would perform with the C-M ordering and node IDs. As they tend to make GNNs more powerful. Actually a recent paper (https://arxiv.org/pdf/2307.01646.pdf) showed quite some improvements in the one-shot graph generators by using a known ordering. It would make sense to include this in the baselines.

**Questions:**

I would really like to see the uniqueness and novelty for all the models in the datasets that were tested. Validity as introduced in SPECTRE is also an interesting measure to have, that's maybe more easy to interpret than the MMDs, where it is available.

I'll raise my score if this is addressed. I just don't think we can accept the paper without this information.

### After Rebuttal
I thank the authors for all the additional experiments and adjustments, esp. w.r.t. BwR.
While the novelty and uniqueness of generated graphs is mediocre and performance is not really better than the newest one-shot models (e.g. SwinGNN), the proposed change is neat and does improve upon autoregressive baselines. Thus I raise my score.

---

> ### Author Response · Authors · 2023-11-17
>
> Dear reviewer EmZF,
>
> We sincerely appreciate your comments and efforts in reviewing our paper. We think our paper has much improved in terms of comprehensive evaluation and clarity, thanks to your comment. We especially found your comment **W1** very insightful!
>
> We address your question as follows. We also updated our manuscript which is highlighted in $\color{red}{\text{red}}$.
>
> ---
>
> **W1/Q1. The novelty, uniqueness, and validity of the generated examples are not reported.**
>
> Thank you for such an insightful comment! Following your suggestion, we evaluated the validity, novelty, and uniqueness of GEEL and the considered baselines. We report the new results in Table 15 and Table 16 of Appendix F. We think our empirical results are now much more comprehensive and reliable, thanks to your review.
>
> In our new experiments, GEEL achieves around 30\% \~ 90\% novelty, which is better than autoregressive models like GRAN and BiGG, but lower than the diffusion-based graph generative models as the reviewer anticipated. This is partially due to the inherent trade-off between novelty and the capability of generative models to faithfully learn the data distribution. Even diffusion models suffer from this trade-off, e.g., DiGress achieves around 30\% novelty for the QM9 dataset. We also note that this trade-off is less severe for harder problems, e.g., point cloud and ZINC250k, which is the main target of our work.
>
> Finally, we note the minor modification of GEEL in the new experiments. Our GEEL now samples a new C-M ordering for a graph at each training step (instead of fixing a unique ordering per graph). We found this to improve novelty without a decrease in performance and changing the architecture. We updated all of our experimental results to incorporate this change. Note that the results for the Ego dataset are to be updated, as the experiments have not yet been completed.
>
> ---
>
> **W2. A recent paper (SwinGNN) showed quite some improvements in the one-shot graph generators by using a known ordering.**
>
> Thank you for the suggestion. We provide the comparison below and in our updated manuscript.
>
> |         |           | Ego-small |           |           | Com-small |           |
> | ------- |:---------:|:---------:|:---------:|:---------:|:---------:|:---------:|
> |         |   Deg.    |   Clus.   |  Orbit.   |   Deg.    |   Clus.   |  Orbit.   |
> | SwinGNN | 0.000 | 0.021 | 0.004 | 0.003 | 0.051 | 0.004 |
> | GEEL (Ours)    | 0.020 | 0.035 | 0.012 | 0.020 | 0.022 |   0.006   |
>
> |         |       | Grid |        |       | Proteins |        |
> | ------- |:-----:|:---------:|:------:|:-----:|:---------:|:------:|
> |         | Deg.  |   Clus.   | Orbit. | Deg.  |   Clus.   | Orbit. |
> | SwinGNN | 0.000 |   0.000   | 0.000  | 0.002 |   0.016   | 0.003  |
> | GEEL (Ours)    |   0.000    |  0.000         |     0.000   |    0.002   |   0.001        |    0.004    |
>
> Note that SwinGNN does not provide any novelty or uniqueness, hence it is unclear whether if SwinGNN bypasses the overfitting issue. For the considered metrics, both GEEL and SwinGNN showed high graph generation quality.

---

> ### Author Response · Authors · 2023-11-21
>
> Additionally, we clarify that the usage of C-M ordering for autoregressive graph generation is first proposed by BwR [1], not by us. To avoid confusion, we updated our manuscript to clarify this.
>
> [1] Diamant et al., Improving Graph Generation by Restricting Graph Bandwidth, ICML 2023.

---

> > ### Author Response · Authors · 2023-11-23
> >
> > Dear reviewer EmZF,
> >
> > Thank you for your time and efforts in reviewing this paper. Since the discussion phase is close to the end, we would like to inquire if our responses have addressed your concerns.
> >
> > We are especially curious since you highlighted the importance of novelty and uniqueness metrics for raising the score. We wonder if our response has resolved the issue.
> >
> > We remain fully committed to addressing any questions you may have by the end of the discussion phase.

---

### Public Comment · ~Nathaniel_L._Diamant1 · 2023-11-17
**Concerns around lack of attribution to prior work**

Dear Area Chairs and Reviewers,

We would like to bring to your attention that this work potentially misrepresents its contributions, bringing into question its novelty. The use of Cuthill-Mckee orderings, bandwidth restriction in auto-regressive models (and one-shot and diffusion models), and the observation of the low bandwidth of molecular graphs, were all first introduced in our work: Diamant et al. 2023, _Improving Graph Generation by Restricting Graph Bandwidth_ (or “BwR”), [published at ICML](https://proceedings.mlr.press/v202/diamant23a). We note that most of the supplementary code relies on our [open source repository](https://github.com/Genentech/bandwidth-graph-generation) for bandwidth ordering (e.g. data/orderings.py).

In particular, we would like to highlight several concerns:

1. __Lack of proper attribution to prior work__. This manuscript presents multiple statements without attribution that give the impression they are part of the paper's contributions. For example:

    * "Applying bandwidth restriction" is mentioned as a key contribution without citation to our method, BwR [1], which was the first to propose bandwidth-restricted orderings for graph generation.

    * Using a recurrent neural network with positional encodings to generate a low bandwidth representation is listed as a key contribution, again without citation to our work. BwR did the same when applied to GraphRNN, resulting in asymptomatic runtime in O(NB) compared to GEEL's O(MB).

    * The first paragraph of page two reads, "We also promote the use of C-M ordering", without citing our work, whose core contributions heavily featured using the C-M ordering for graph generation.

    *  On page four, the authors say, without citation, that "many real-world graphs, such as molecules and community graphs, exhibit low bandwidths as shown in Appendix C." This is listed as the first contribution of our work [1], which reads: "We show that many real-world classes of graphs, such as molecules, have low graph bandwidth." Prior to our publication, this was not a well-established fact.

2. __Novelty__ This manuscript relies on the bandwidth parametrization as a core component of the method. Given that such parametrization is a separate method proposed in [1], the authors should introduce and evaluate its impact separately.
    * BwR [1] first proposed and applied a bandwidth parametrization to several graph generative models (including an autoregressive model), and proposed itself as a general-purpose method that “_can be applied to virtually all existing graph generation methods_” [1]. In this context, the combination of bandwidth parametrization with an edge list-based representation proposed by GEEL appears to be an application of BwR to edge-list-based graph generation.
    * As highlighted by the authors in the Related Works (page 3), “a few works have presented edge list-based representations”, and the main contribution of GEEL appears to be the more compact representation used, which is achieved through the gap encoding and the bandwidth parametrization. However, the latter is exactly the main contribution of BwR [1]. The authors should better clarify the novelty and main contribution of GEEL and its impact in isolation compared to previous edge list-based representations and in conjunction with BwR.

Continued in following comment.

---

> ### Public Comment · ~Nathaniel_L._Diamant1 · 2023-11-17
> **Concerns around lack of attribution to prior work continued**
>
> 3. __Lack of comparisons to prior work and opaque experiments.__
>     * Table 1 (generative performance) includes BwR without explaining which graph generative model BwR was applied to. BwR is a general framework to constrain the bandwidth of graph generative models that has been applied to autoregressive, VAE-based, and score-based methods in prior work [1]; it is not a model in itself.
>     * This lack of explanation of how BwR was used is mirrored in the Appendix, which does not explain which generative model BwR was tested with. This means the results are not reproducible from the paper's text.
>     * Table 1 (generative performance) has missing values for some model/dataset combinations (in particular, BwR). Additionally, Table 13 (additional experimental results) does not include BwR.
>     * Table 2 (inference times) misleadingly leaves out BwR, which would be the natural comparison. For reference, Table 4 of Diamant et al. 2023 [1] shows how bandwidth restriction can improve inference time for nearly any graph generative modeling approach. The authors should consider quantifying the speed-up given by their proposed model without using BwR, and comparing it to the speed-up given by BwR.
>    * The authors include an ablation study using different orderings (Section 4.3, Figure 5). However, these results only include training curves and not the impact on the generative performance (which would reflect the contribution of BwR to the method’s performance). The authors should consider separately evaluating their proposed model and the impact of adding BwR on top of it.
>
> In conclusion, the authors designed GEEL by applying BwR to autoregressive edge-list generation, and used the BwR code to do so. The relevance of graph bandwidth and Cuthill-Mckee to molecular graphs was also an insight of Diamant et al. 2023. If the work were framed as focusing on the edge-gap encoding and applying BwR, we believe it would be a much more accurate representation of its novelty. In that case, the authors should separately introduce and evaluate the impact of the edge-gap encoding and potentially show synergies with the previously-introduced bandwidth parametrization. As is, the work claims many insights that are not its own.
>
> We thank the reviewers for their time and consideration,
> Nathaniel Diamant, Alex M. Tseng, Kangway V. Chuang, Tommaso Biancalani, Gabriele Scalia
>
> [1]  Diamant et al., _Improving Graph Generation by Restricting Graph Bandwidth_, ICML 2023.

---

> ### Author Response · Authors · 2023-11-18
>
> Thank you for the comments and for expressing your concern about our work. We are very disappointed and sorry to hear that our work does not put proper attribution on BwR, despite referring to it in the related work section. We sincerely hope to fix this issue regardless of the evaluation of our work.
>
> Incorporating your concerns, we would like to clarify our contribution as follows: *Our work proposes a new edge list representation with the vocabulary size of $B^2$ with gap encoding. We promote reducing the bandwidth $B$ using the C-M ordering, **as proposed by BwR**.* Please note that our primary contribution, i.e., the gap encoding edge list, is orthogonal to BwR, since one could also consider the BFS ordering (used by GraphRNN) to reduce the vocabulary size $B^2$.
>
> Importantly, most of the concerns seem to arise from our initial conception that BwR is about restricting the bandwidth for "reduction of the ***adjacency matrix representation*** for graph generation", hence not directly related to our edge list representation. This was partly due to the BwR description being focused on the adjacency matrix-based graph generative models. Now we would like to incorporate your viewpoint that "*BwR is about constraining **any graph generative model** via the C-M node ordering*".
>
> To alleviate your concern, we revised the paper to fully attribute to your work *whenever* we use the C-M ordering. Our main updates (marked in $\color{magenta}{\text{magenta}}$) are as follows:
> - Instead of promoting the bandwidth restriction *via C-M ordering*, we promote the bandwidth restriction *via BwR*.
> - We updated Table 1 and 2 to incorporate all the BwR baselines.
> - We will soon update other missing experiments that you pointed out.
>
> In what follows, we reply to your comments one by one.
>
> **0. Most of the supplementary code for bandwidth ordering relies on BwR open-source repository.**
>
> Please note that we already included references to the public repository of BwR in our supplementary code. To make this more clear, we additionally added the code references in Appendix B.4. in the updated manuscript. If you are uncomfortable with this, we would be happy to implement our own C-M ordering algorithm.
>
> **1-1. "Applying bandwidth restriction" is mentioned as a key contribution without citation to our method, BwR.**
>
> We clarify that we mentioned that *"BwR proposed to constrain the bandwidth via C-M ordering, bypassing the generation of out-of-bandwidth elements, which reduces the representation size to $NB$"* in the second paragraph of Related works. Moreover, in our updated manuscript, we clarify that we fully attributed your BwR for reducing the maximum bandwidth using the C-M node ordering.
>
> **1-2. The first paragraph of page two reads, "We also promote the use of C-M ordering".**
>
> We clarified this to "We promote the use of C-M ordering as proposed by BwR".
>
> **1-3. Using a low bandwidth representation is listed as a key contribution, again without citation to our work. BwR did the same when applied to GraphRNN, resulting in asymptomatic runtime in O(NB) compared to GEEL's O(MB).**
>
> Using the gap-encoded edge list representation with vocabulary size (not representation size) bounded by the bandwidth is indeed our contribution. In our updated manuscript, we now refer to your work by stating that "Our representation further reduces the vocabulary size using the BwR scheme."
>
> We also note that the number of iterations required for generating the representations is $M$ (not $O(MB)$) for GEEL while it is $NB$ for GraphRNN+BwR. The length of our representation is fixed regardless of the node ordering or the bandwidth.
>
> **1-4. The authors say, without citation, that "many real-world graphs exhibit low bandwidths as shown in Appendix C."**
>
> In our updated manuscript, we referred to your BwR on finding out real-world graphs have low bandwidth. Note that we do not claim to be the first to discover this fact.
>
> **2-1. The combination of bandwidth parameterization with an edge list-based representation proposed by GEEL appears to be an application of BwR to edge-list-based graph generation.**
>
> With all due respect, we disagree that GEEL is an application of BwR. While we now clarify that our C-M ordering stems from existing works including BwR in the updated manuscript, our new gap encoding (which is our main contribution) is not related to any concepts proposed by BwR. Our analysis of $B^2$ vocabulary size simply follows from the definition of graph bandwidth.
>
> **2-2. The authors should better clarify the novelty and main contribution of GEEL and its impact in isolation compared to previous edge list-based representations and in conjunction with BwR.**
>
> The main contribution of GEEL with respect to both edge list-representations is the gap-encoded edge list representation, of which the vocabulary size can be further reduced by the BwR scheme.

---

> > ### Author Response · Authors · 2023-11-18
> >
> > **3-1, 3-2, 3-3. The experiments do not compare with BwR in an opaque way.**
> >
> > To alleviate your concern, we added the three versions of BwR in our updated manuscript: BwR + GraphRNN, BwR + Graphite, and BwR + EDP-GNN. Note that the originally reported version was BwR + GraphRNN, which showed the best MMD scores among the three versions. In addition, Table 13 does not include BwR as the baseline since the performance for Ego-small and Community-small datasets we used were missing in the BwR paper. Finally, in our updated manuscript, we clarified our description on which version of BwR was used for comparison.
> >
> > **3-4. Table 2 misleadingly leaves out BwR, which would be the natural comparison. The authors should consider quantifying the speed-up given by their proposed model without using BwR, and comparing it to the speed-up given by BwR.**
> >
> > We will soon report the inference time of our GEEL with and without using BwR to compress the vocabulary (before end of the rebuttal phase). Note that we anticipate the difference to be small since BwR reduces the memory requirement, not length of the representation, for the LSTM.
> >
> > **3-5. The authors include an ablation study using different orderings (Section 4.3, Figure 5). However, these results only include training curves and not the impact on the generative performance (which would reflect the contribution of BwR to the method’s performance).**
> >
> > We have not provided the generation performance as generation with any ordering eventually converges to the same levels of MMD, e.g., the average MMD of 0.000 for Grid graphs. We now mention this in our updated manuscript.
> >
> > **Conclusion. The authors designed GEEL by applying BwR to autoregressive edge-list generation, and used the BwR code to do so. The relevance of graph bandwidth and Cuthill-Mckee to molecular graphs was also an insight of Diamant et al. 2023. The authors should separately introduce and evaluate the impact of the edge-gap encoding and potentially show synergies with the previously-introduced bandwidth parametrization.**
> >
> > We think it is hard to conclude that GEEL is the application of BwR to edge list representation since our main contribution, i.e., gap encoding, cannot be derived from BwR. We made our manuscript to be more clear on how (1) we use the BwR code for C-M ordering as explained in Appendix B.4. and (2) are not the first to discover the useful-ness of C-M node ordering for graph generation. We believe our work sufficiently introduces and evaluates the edge-gap encoding (in Table 1, 4, 6) and show synergies with the different node orderings (Figure 5).
> >
> > In conclusion, we resonate with your concern that GEEL does not put sufficient attributions to the insights discovered by BwR. However, this is mainly due to our unclear presentation and *we did not make any false claim that we discovered the insights by ourselves*. This is now clearly stated in our revised manuscript.

---

### Author Response · Authors · 2023-11-21

Dear reviewers (**aPij**, **RDfu**, **kVSU**, and **EmZF**) and area chairs,

We are deeply grateful for the time and effort you spent reviewing our manuscript. As the rebuttal phase is drawing to a close, **we would greatly appreciate the reviewers for checking our response.**

Importantly, we sincerely request the reviewers and the area chairs to check the concerns expressed by the authors of BwR. It is worth noting that our GEEL is not a variant of BwR, i.e., our main contribution, gap encoding of edge tokens is not related to any concepts proposed by BwR. We believe our response and the updated manuscript alleviate the concerns, but we would be grateful for more discussion on this issue.

Aside from this issue, we are heartened by the reviewers’ positive reception of our work, particularly in regards to the novel idea (**kVSU, RDfu**), the compactness (**aPij, kVSU, EmZF**), the strong empirical results (**EmZF, kVSU**), and the clarity of presentation (**EmZF, kVSU, RDfu**).

Additionally, we also point out important concerns highlighted by the reviewers, which we have responded to alleviate the concerns.

Reviewer **aPij** questioned on the fairness of the inference time analysis using different GPUs. We clarified that we actually used the same GPU for all models in the inference time analysis.

Reviewers **EmZF** and **RDfu** pointed out the missing novelty and uniqueness metrics for general graph generation. We reported the missing metrics, where GEEL achieved better novelty than other autoregressive baselines.

Reviewers **aPij** and **kVSU** pointed out the missing novelty and uniqueness metrics for molecular graph generation. We reported the missing metrics, where GEEL showed competitive novelty and uniqueness.

Finally, we summarize the revisions of our updated manuscript.

- Clarification of the fair setting of inference time analysis
- Clarification of the attribution to prior work
- Additional experimental results on novelty and uniqueness for general and molecular graph generation
- Additional experimental results on the MOSES benchmark
- Discussion on the limitations of our work
- Incorporation of all the editorial comments

Thank you very much for your valuable time and effort,

Authors

---

### Meta-Review · Area_Chair_CeRy · 2023-12-05

**Metareview:**

This paper proposes a novel edge-list based graph representation which allows for enhanced performance in graph generation. All reviewers recognise the utility of the work in this regard. While the assessment is not unanimously in favour of acceptance, I see a clear sign that the paper is over the bar for ICLR, and there was no opposition from the countering Reviewer. I wholeheartedly recommend acceptance.

**Justification For Why Not Higher Score:**

While the gains in computational performance are clearly observed, doubts were raised about how such gains would transfer to practical impact, for example in drug design problems. Without a clear indication of downstream impact and utility, I do not think this work should be nominated for a spotlight.

**Justification For Why Not Lower Score:**

The authors present a compelling contribution and have very carefully handled the discussion period---including responding to non-reviewer comments in an appropriate way. This is a clear accept in my view.

---

### Decision · Program_Chairs · 2024-01-16

Accept (poster)